# Autophagy-deficient macrophages exacerbate cisplatin-induced mitochondrial dysfunction and kidney injury via miR-195a-5p-SIRT3 axis

Yujia Yuan[1,5], Longhui Yuan[1,5], Jingchao Yang[1], Fei Liu[1,2], Shuyun Liu[1], Lan Li[1], Guangneng Liao[3], Xi Tang[4], Jingqiu Cheng[1], Jingping Liu [1], Younan Chen[1,2] ✉ & Yanrong Lu [1] ✉

Macrophages (Mφ) autophagy is a pivotal contributor to inflammation-related diseases. However, the mechanistic details of its direct role in acute kidney injury (AKI) were unclear. Here, we show that Mφ promote AKI progression via crosstalk with tubular epithelial cells (TECs), and autophagy of Mφ was activated and then inhibited in cisplatin-induced AKI mice. Mφ-specific depletion of ATG7 (Atg7^Δmye) aggravated kidney injury in AKI mice, which was associated with tubulointerstitial inflammation. Moreover, Mφ-derived exosomes from Atg7^Δmye mice impaired TEC mitochondria in vitro, which may be attributable to miR-195a-5p enrichment in exosomes and its interaction with SIRT3 in TECs. Consistently, either miR-195a-5p inhibition or SIRT3 overexpression improved mitochondrial bioenergetics and renal function in vivo. Finally, adoptive transfer of Mφ from AKI mice to Mφ-depleted mice promotes the kidney injury response to cisplatin, which is alleviated when Mφ autophagy is activated with trehalose. We conclude that exosomal miR-195a-5p mediate the communication between autophagy-deficient Mφ and TECs, leading to impaired mitochondrial biogenetic in TECs and subsequent exacerbation of kidney injury in AKI mice via miR-195a-5p-SIRT3 axis.

Acute kidney injury (AKI) is a common and severe clinical complication in critically ill patients and is characterized by inflammatory tubular damage and the consequent reversible loss of renal function[1,2]. Recent epidemiologic studies have reported that the incidence of AKI is as high as 10–15% in hospitalized patients and 50% in intensive care patients, which has become a global health concern[3].

Tubulointerstitial inflammation is the premonitory pathology in AKI and involves interactions among multiple immune cells, including macrophages (Mφ), neutrophils, dendritic cells (DCs), and T cells[4]. Mφ play a vital role in driving the progression of AKI, depending on their activation[5]. Autophagy is considered a catabolic degradation process for maintaining intracellular homeostasis[6]. Liu et al. demonstrated that the inhibition of autophagy could skew Mφ polarization to a proinflammatory M1 phenotype[7], and increasing evidence suggests that Mφ autophagy is the underlying mechanism element responsible for inflammation-related diseases, including sepsis[8], atherosclerosis[9], and liver diseases[10,11]. However, Hu et al. reported that complement C5a can activate alveolar Mφ autophagy,

[1]National Health Commission (NHC) Key Laboratory of Transplant Engineering and Immunology, West China Hospital, Sichuan University, Chengdu, China. [2]Institutes for Systems Genetics, West China Hospital, Sichuan University, Chengdu, China. [3]Animal Center, West China Hospital, Sichuan University, Chengdu, China. [4]Department of Nephrology, West China Hospital, Sichuan University, Chengdu, China. [5]These authors contributed equally: Yujia Yuan, Longhui Yuan. ✉e-mail: chenyounan@scu.edu.cn; luyanrong@scu.edu.cn

leading to cell apoptosis and the destruction of pulmonary micro-environment homeostasis, which further aggravates acute lung injury[12]. In addition, elevated autophagy can induce Mφ polarization to the M1 phenotype, which impairs cutaneous wound healing in diabetic mice[13], indicating that the regulatory effects of autophagy on Mφ polarization and its role in different disease processes are inconsistent. Therefore, limited studies have been conducted to identify the association between Mφ autophagy and kidney disease. Recently, one study revealed that loss of mitophagy proteins PINK1 or Parkin aggravated the Mφ-derived fibrotic response and kidney fibrosis[14]. However, the direct role of Mφ autophagy in the development of AKI and the underlying mechanism are still unclear.

Crosstalk between Mφ and tubular epithelial cells (TECs) is essential for the progression of AKI[3,15]. Exosomes are a subtype of secreted vesicle that can mediate communication via cell-specific cargo, including nucleic acids, proteins, and metabolites, to recipient cells[16]. Several studies have confirmed that exosomal transfer of cargo from TECs to Mφ constitutes a critical mechanism of albumin or LPS-induced tubulointerstitial inflammation[17–19]. In addition to exosomal transfer from TECs to Mφ, elevated exosomal miR-93 derived from M2 Mφ inhibited pyroptosis pathway in epithelial cells and alleviated kidney injury in AKI mice, indicating that exosomal transfer between TECs and Mφ is bidirectional[20]. Overall, these findings suggested that exosomal transfer-mediated crosstalk between Mφ and TECs might be involved in the Mφ autophagy mediated regulation of kidney injury in AKI mice.

Here, we report the mechanism by which Mφ autophagy mediates the regulation of kidney injury in AKI mice. Using genetically engineered mice with myeloid cell-specific deletion of ATG7, we observe autophagy-deficient Mφ induced systemic inflammation, impaired mitochondria, and aggravated kidney injury in AKI mice. Mechanistically, we found that autophagy-deficient Mφ-derived exosomes impaired mitochondria in TECs via miR-195a-5p and that miR-195a-5p impaired mitochondria in Atg7$^{\Delta mye}$ mice via targeting SIRT3. Furthermore, we show that in Atg7$^{\Delta mye}$ mice, targeting miR-195a-5p/SIRT3 axis alleviated cisplatin-induced kidney injury. Our findings indicate that Mφ autophagy deficiency exacerbates kidney injury via miR-195a-5p/SIRT3 axis and Mφ autophagy emerges as a potential therapeutic target in AKI and other inflammation-related diseases.

## Results

### Mφ autophagy was activated first and then suppressed in AKI mice

Kidney injury was induced by cisplatin injection, as characterized by increased blood urea nitrogen (BUN) and creatinine (CREA) levels (Fig. S1a, b) and the abnormalities in renal structure, compared with those in normal control (NC) mice (Fig. S1c, d). We also observed a striking increase in the serum levels of the proinflammatory factors TNF-α and IL-1β (Fig. S1e, f), indicating that systemic inflammation was caused by cisplatin.

Mφ are key mediators of inflammation, and Mφ infiltration in the kidneys of AKI mice was measured on the 2nd, 3rd, and 4th days after cisplatin injection. We found that Mφ accumulation substantially increased in a time-dependent manner in AKI mice (Fig. 1a). We first determined changes in Mφ autophagy in cisplatin-induced AKI. Compared with those in NC mice, the levels of LC3 II, ATG7, and BECN1 in the peritoneal Mφ were increased on the 2nd day after cisplatin injection and then quickly decreased to levels below those in than NC mice, while the protein level of P62 increased (Fig. 1b, c). Similarly, a dramatic decrease in the percentage of Mφ with LC3-II puncta was observed on the 4th day after cisplatin treatment (Fig. 1d). The expression of LC3 II could be influenced by the maturation of autophagosomes and autophagic flux. To further analyze the cause of decreased LC3 II expression, Mφ were treated with the lysosomal inhibitor hydroxychloroquine (HCQ), which inhibits autophagic

hyperactivity by blocking autophagosome fusion and degradation. According to the guidelines for the use and interpretation of assays for monitoring autophagy (4th edition)[21], the autophagic flux index (defined as the proportion of the number of LC3-II or LC3 puncta in the presence of HCQ to that in the absence of HCQ) was calculated at the indicated times. With HCQ treatment, the fold change in the number of LC3 puncta in Mφ from AKI mice (Mφ$^{AKI}$) was less than that in NC mice, indicating that autophagic flux was inhibited in Mφ from AKI mice (Fig. 1d). In addition, after HCQ treatment was used to inhibit autophagic flux activity, LC3 immunofluorescence reflected the relative rate of autophagosome formation. Our results showed that the MFI of LC3-treated Mφ$^{AKI}$ cells was lower compared with that of Mφ$^{NC}$ cells in HCQ group, suggesting that autophagosome formation was reduced in Mφ$^{AKI}$ (Fig. 1d). To detect the autophagy activity of kidney-infiltrated Mφ in AKI mice, Mφ labeled with F4/80 were sorted from the kidney, and we found that the expression of LC3 decreased significantly after an initial increase in AKI mice (Fig. 1e). Consistently, immunofluorescence staining of LC3 in kidney-infiltrated Mφ on the 4th day after cisplatin treatment decreased relative to that in NC mice (Fig. 1f). Overall, these findings confirm that Mφ autophagy in AKI mice is activated first and then inhibited.

### Autophagy deficiency in Mφ exacerbated inflammation in AKI mice

To investigate the role of Mφ autophagy in cisplatin-induced AKI, we generated myeloid cell-specific ATG7-deficient (Atg7$^{\Delta mye}$) mice (Fig. S2a, b). To confirm LysM-Cre mediated recombination in our research, we measured the expression of autophagy-related genes in kidney Mφ (KM) and peritoneal Mφ (PM). The results showed that LysM-Cre-mediated recombination was highly efficient in the KM and PM of the Atg7$^{\Delta mye}$ mice (Fig. S2c–e). After cisplatin injection, the number of kidney-infiltrated Mφ in the Atg7$^{\Delta mye}$ mice was greater than that in the WT mice (Fig. 2a). Moreover, a significant increase in the M1 Mφ population in the kidney tissues of the Atg7$^{\Delta mye}$ mice was observed after cisplatin injection (Fig. 2b, c). TNF-α in kidney tissue was significantly upregulated in the Atg7$^{\Delta mye}$ mice (Fig. 2d). Moreover, the mRNA and protein levels of proinflammatory factors were also increased (Fig. 2e, f), indicating that Mφ deletion of ATG7 promoted systemic inflammation.

### Autophagy deficiency in Mφ sensitized mice to cisplatin-induced AKI

Compared to that in WT mice, more severe renal dysfunction was observed in Atg7$^{\Delta mye}$ mice, demonstrated by higher levels of BUN and CREA (Fig. 3a, b). Histology analysis verified that cisplatin-induced notable tissue damage, including tubular necrosis, dilatation, vacuolization, and detachment (Fig. 3c). In addition, the area of immunostained kidney injury molecule 1 (KIM1) was significantly enlarged in Atg7$^{\Delta mye}$ mice (Fig. 3c), which was accompanied by marked increases in KIM1 mRNA and neutrophil gelatinase-associated lipocalin (NGAL) mRNA (Fig. 3d). Moreover, TUNEL+ cells were substantially more abundant in the kidneys of the Atg7$^{\Delta mye}$ mice than in those of the WT mice after cisplatin injection (Fig. 3e). Similarly, along with the increased expression of BAX, the Bcl-2 level was reduced in the cisplatin-treated Atg7$^{\Delta mye}$ mice (Fig. 3f). To better understand the specific role of macrophages (Mφ) in injury during AKI, we conducted experiments involving Mφ depletion in cisplatin-treated Atg7$^{\Delta mye}$ mice, with a particular focus on autophagy. After clodronate liposome (Lipo-Clod) injection, kidney-infiltrated Mφ were significantly depleted, as determined by flow cytometry (Fig. S3a). Compared to those in cisplatin-induced Atg7$^{\Delta mye}$ mice, the serum BUN and CREA levels were decreased in Lipo-Clod-treated mice (Fig. S3b, c). Kidney morphological abnormalities and tubular damage scores were ameliorated after Mφ depletion (Fig. S3d), indicating that Mφ depletion protected against cisplatin-induced kidney injury in Atg7$^{\Delta mye}$ mice. Collectively,

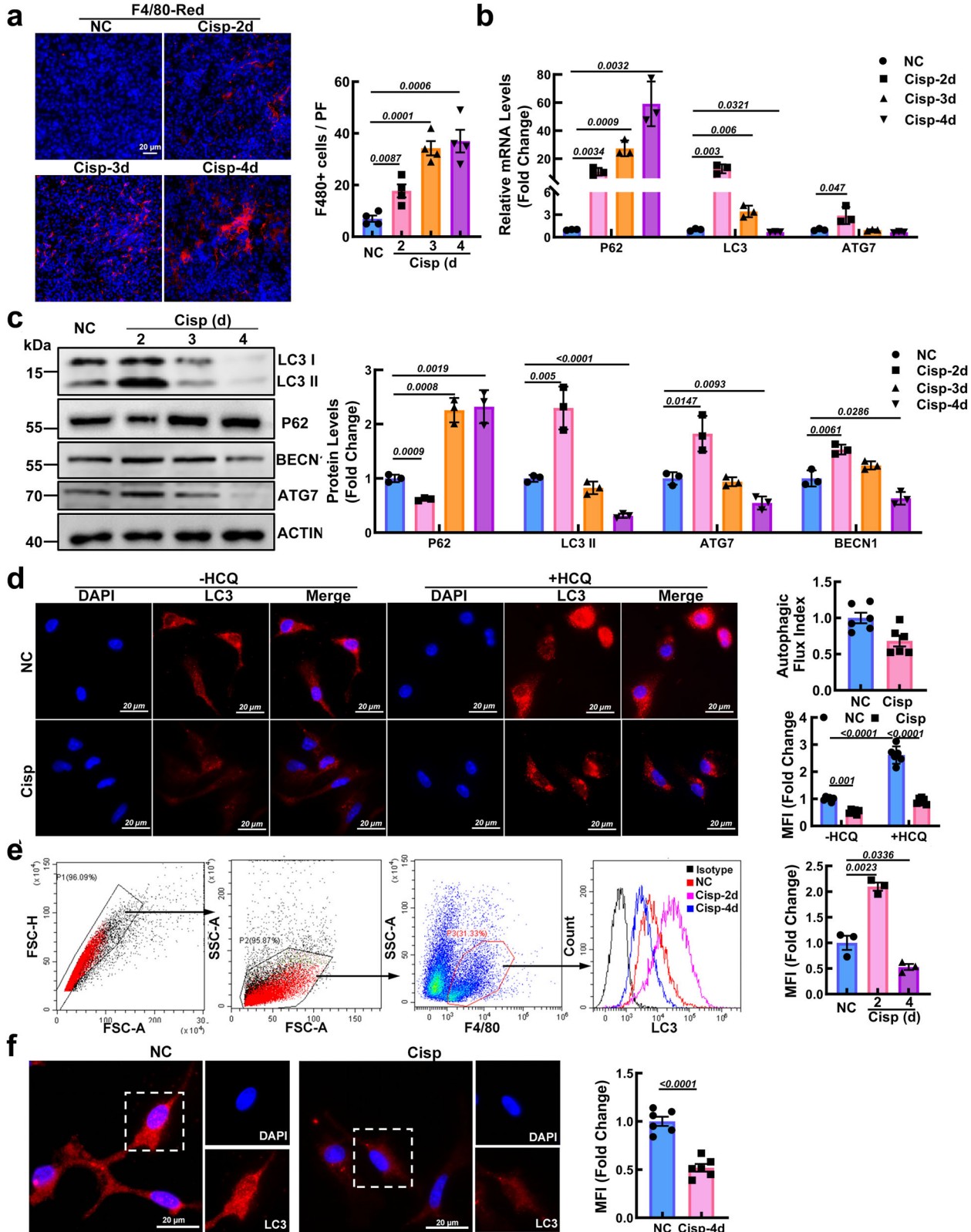

these results suggest that autophagy deficiency in Mφ aggravates kidney injury and apoptosis in AKI mice.

## Exosomes derived from Mφ in Atg7$^{\Delta mye}$ mice contribute to the aggravated kidney injury

To evaluate whether exosome-mediated crosstalk between Mφ and TECs affects kidney injury, Mφ-derived exosomes from mice (Mφ$^{WT}$-

EXO, EXO1; Mφ$^{Atg7\Delta mye}$-EXO, EXO2) were isolated and characterized by TEM, NTA, and western blotting (Fig. S4a–d). The results showed that exosome-sized particles did not significantly change in EXO1 or EXO2 in the presence of equivalent amounts of Mφ$^{WT}$ or Mφ$^{Atg7\Delta mye}$, respectively (Fig. S4e). The total exosomal protein content per million cells was greater in the EXO2 group than in the EXO1 group (Fig. S4f). To explore the uptake of exosomes in vitro, TECs were cultured with

**Fig. 1 | Autophagy activity of macrophages (Mφ) is activated and then inhibited in cisplatin-induced AKI mice. a** Representative images of F4/80 (red) in kidney sections used for Mφ detection. Scale bar, 20 μm, $n = 4$ biological replicates for each group, unpaired two-tailed Student's t test. **b, c** After cisplatin injection, peritoneal Mφ were isolated, and the mRNA levels (**b**) (unpaired two-tailed Student's t test) and protein levels (**c**) (one-way ANOVA with Dunnett's multiple comparisons test) of autophagy genes were measured. ACTIN was used as the loading control. $n = 3$ biological replicates for each group. **d** Representative images of LC3 staining (red) in peritoneal Mφ treated with the treatment of hydroxy-chloroquine (HCQ, 20 μM) after cisplatin injection for 4 days. Scale bar, 20 μm, $n = 6$ biological replicates for each group, two-way ANOVA with Tukey's multiple comparison test. To detect changes in autophagy in kidney-infiltrated Mφ, F4/80+ cells in kidney tissue were sorted. **e** Representative flow cytometric images illustrating the expression of LC3 in F4/80+ cells in kidney tissues. $n = 3$ biological replicates for each group, unpaired two-tailed Student's t test. **f** F4/80+ kidney-infiltrated Mφ on the 4th day after cisplatin treatment were sorted and then stained with LC3 (red). Scale bar, 20 μm, $n = 6$ biological replicates for each group, unpaired two-tailed Student's t test. The data are presented as the means ± SEMs. NC normal control, Cisp cisplatin. Source data are provided as a Source data file.

PKH67-labeled exosomes for 24 h. The fluorescence of internalized exosomes in TECs was found surrounded the nucleus or lined the inner surface of the cell membrane (Fig. S4g). As result of exosome-mediated direct communication, the viability of TECs decreased in a concentration-dependent manner in response to EXO2, while apoptosis increased after EXO2 treatment (Fig. S4h, i).

To explore the role of Mφ-derived exosomes in intercellular communication in vivo, exosomes were labeled with DiR (Fig. 4a). The fluorescence tracer demonstrated that the exosomes were mainly distributed in the liver, spleen, lung, heart and kidney (Fig. 4b). Consistent with a previous study showing that integrin-mediated adhesive interactions guide the homing of Mφ-derived exosomes to inflamed kidneys[22], we found that the renal radiance signals in AKI mice were significantly greater than those in NC mice, suggesting that Mφ-derived exosomes exhibit preferential tropism to injured kidneys (Fig. 4b, c). Consequently, renal dysfunction and destruction of renal structure in AKI mice were obviously increased with EXO2 treatment (Fig. 4d–f). In contrast, EXO1 had minimal effects on renal function and the severity of AKI, suggesting that Mφ-derived exosomes from Atg7$^{\Delta mye}$ mice exacerbated kidney injury.

## Exosomes derived from autophagy-deficient Mφ impaired mitochondria in TECs via miR-195a-5p

Exosomes contain a variety of cargoes, including miRNAs, other RNA species, lipids, and proteins. Previously, based on the miRNA profile, we demonstrated that miR-195a-5p in Mφ-derived exosomes from AKI mice was increased, while blockade of miR-195a-5p inhibited mitochondrial damage in AKI mice[23]. Here, the level of miR-195a-5p in Mφ from Atg7$^{\Delta mye}$ mice was 55.08 ± 6.34 fold higher than that in Mφ from WT mice and was further elevated after cisplatin injection (Fig. 5a). Moreover, the miR-195a-5p level was also higher in EXO2 than in EXO1 (Fig. 5b). Due to miRNA transfer by exosomes, a higher level of miR-195a-5p in recipient TECs was also measured after EXO2 incubation in vitro (Fig. 5c). Similar to the findings of a previous study, the mitochondrial depolarization and excessive generation of mtROS induced by EXO2 were alleviated by the miR-195a-5p inhibitor (Fig. 5d, e). Mitochondrial fragmentation is associated with increased mtROS. Compared with the elongated networks of mitochondria in control TECs, the mitochondria in EXO2-induced cells exhibited a smaller and more punctate pattern, whereas the mitochondrial fragmentation caused by EXO2 was improved by miR-195a-5p inhibitor treatment (Fig. 5f). Consequently, the reduction in ATP generation and suppression of the OCR induced by EXO2 were partially reversed by the miR-195a-5p inhibitor (Fig. 5g, h). These data suggest that miR-195a-5p inhibitor not only restores the mitochondrial morphology but also effectively improves mitochondrial bioenergetics in TECs.

## The miR-195a-5p antagomir attenuated mitochondrial dysfunction and kidney injury in cisplatin-induced Atg7$^{\Delta mye}$ mice

Consistent with the in vitro observations, mitochondrial damage in the kidneys of Atg7$^{\Delta mye}$ mice was more severe than that in the kidneys of WT mice after cisplatin injection (Fig. S5a–f). Our previous research, similar to that of other investigators, demonstrated that mitophagy in kidney tissue was activated in cisplatin-induced AKI[24], contrast-

induced AKI[25], and sepsis[26] and that further activation of mitophagy could ameliorate mitochondrial damage. Consistent with these results, mitophagy was activated in the kidneys of cisplatin-induced AKI, which was confirmed by the increased protein expression levels of PINK1, Parkin, and LC3 II (Fig. S5g). However, there were no significant differences in mitophagy-related proteins between WT and Atg7$^{\Delta mye}$ mice under AKI conditions, suggesting that mitochondrial dysfunction in cisplatin-induced Atg7$^{\Delta mye}$ mice may be independent of mitophagy (Fig. S5g).

Although there was no significant difference in miR-195a-5p between WT and Atg7$^{\Delta mye}$ mice, a significant increase in miR-195a-5p in Atg7$^{\Delta mye}$ mice was observed after cisplatin injection (Fig. 6a), indicating that autophagy-deficient Mφ may transfer miR-195a-5p to the kidney under AKI conditions. To further confirm the role of miR-195a-5p in mitochondrial dysfunction in vivo, a miR-195a-5p antagomir modified with a 5′-Cy5 florescence probe was administered to Atg7$^{\Delta mye}$ mice through the tail vein 24 h before cisplatin injection. The signal primarily accumulated in the liver, lung, and kidney of the mice, peaking at 48 h and lasting for 96 h post injection (Fig. 6b). We further observed the fluorescence signal of anti-miR-195a-5p-5′Cy5 in the kidneys by preparing kidney tissue sections (Fig. 6c). TEM examination of kidney sections revealed that the cisplatin-induced decrease in mitochondrial morphology in Atg7$^{\Delta mye}$ mice was partially ameliorated by the miR-195a-5p antagomir (Fig. 6d). Moreover, the reduced protein levels of the ETC complexes were increased (Fig. 6e).

We then sought to determine whether the miR-195a-5p antagomir could rescue kidney injury in cisplatin-induced Atg7$^{\Delta mye}$ mice. The results indicated that the elevated renal BUN and CREA levels were reduced (Fig. 6f). In addition, tubular histological damage was alleviated to some extent by the miR-195a-5p antagomir, which was also confirmed by the expression of KIM1 (Fig. 6g). Regarding cellular apoptosis, the number of TUNEL+ cells and the increase in the level of BAX were also reduced (Fig. 6h, i). These data provide evidence that miR-195a-5p is an important contributor to the severe mitochondrial damage and kidney injury induced by cisplatin in Atg7$^{\Delta mye}$ mice.

## miR-195a-5p impaired mitochondria in Atg7$^{\Delta mye}$ mice by targeting SIRT3

To identify target genes of miR-195a-5p, we used a microRNA target prediction system (www.microrna.org). Among the predicted target genes, a putative-binding site of miR-195a-5p was found in the 3′UTR of SIRT3 mRNA, which is essential for mitochondrial homeostasis (Fig. 7a). According to the studies of Yuan[27] and Liu[28], both SIRT1 and Bcl-2 are direct targets of miR-195, respectively. To determine the specific targets of miR-195a-5p in TECs, the cells were treated with different concentrations of the miR-195a-5p mimic for 48 h, after which the expression of SIRT1, SIRT3, and Bcl-2 was measured. Here, the miR-195a-5p mimic had no significant effect on Bcl-2, regardless of the mimic concentration (10 nM) or the higher concentration (50 nM). Moreover, the expression of SIRT1 slightly decreased until a dose of 50 nM was reached. However, the protein level of SIRT3 was significantly decreased in a concentration-dependent manner (Fig. S6), suggesting that miR-195a-5p may specifically target SIRT3 in TECs. We therefore performed luciferase reporter activity assays to validate this

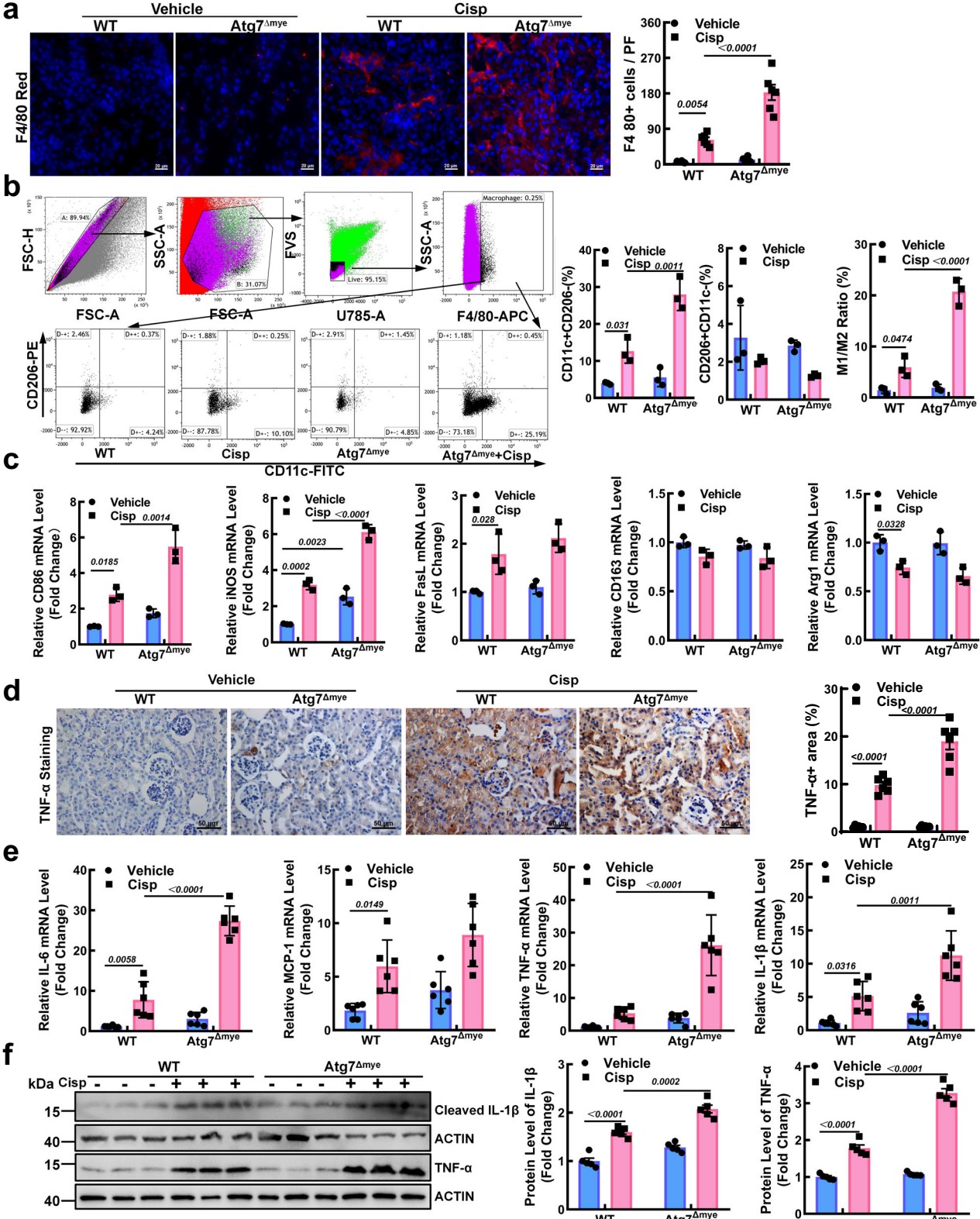

**Fig. 2 | Macrophage (Mφ)-specific deficiency of ATG7 aggravates the inflammatory response in AKI mice. a** Representative images and quantification of F4/80 (red) in kidney sections for Mφ detection. Scale bar, 20 μm, *n* = 6 biological replicates for each group. **b**, **c** The phenotype of Mφ in kidney tissues was characterized by flow cytometry (**b**) (M1: CD11c + CD206-; M2: CD206 + CD11c-) and RT–PCR (**c**) (M1: CD86, iNOS, FasL); M2: CD163 and Arg1). *n* = 6 biological replicates for each group. **d** Immunohistochemical staining for TNF-α in paraffin-embedded kidney sections. Scale bar, 50 μm, *n* = 6 biological replicates for each group. The mRNA levels (**e**) and protein levels (**f**) of inflammation-related genes were measured. ACTIN was used as the loading control. The data are the means ± SEMs. All statistical analysis were performed by two-way ANOVA with Tukey's multiple comparison test. Source data are provided as a Source data file.

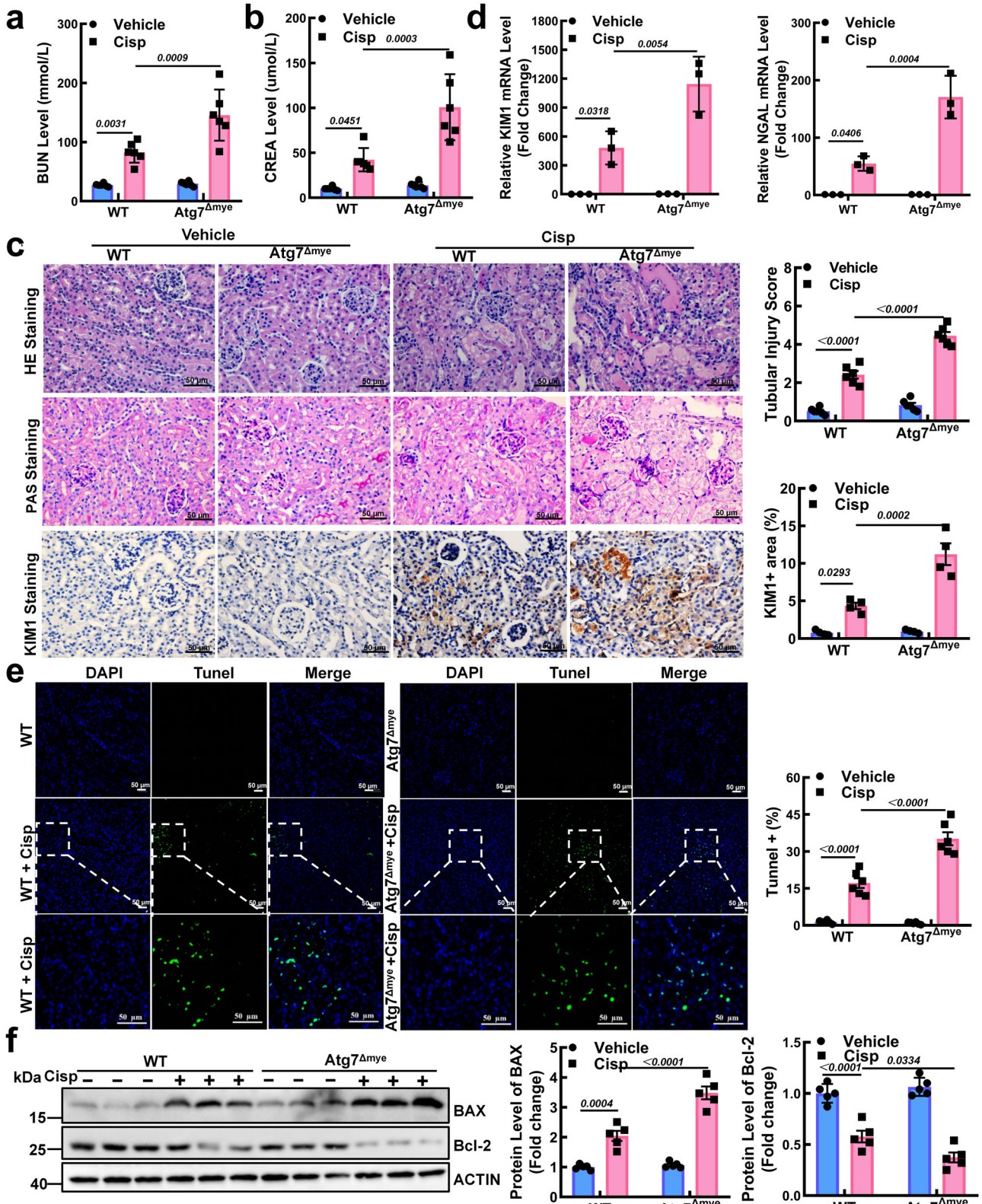

**Fig. 3 | Macrophages (Mφ) specifically deficient in ATG7 sensitize mice to cisplatin-induced kidney injury. a, b** Serum of BUN and CREA in WT and Atg7$^{\Delta mye}$ mice after cisplatin injection. $n = 6$ biological replicates for each group. **c** Representative periodic acid-Schiff (PAS)- and hematoxylin and eosin (H&E)-stained kidneys and immunohistochemical staining and quantification of kidney injury molecule 1 (KIM1) in paraffin-embedded kidney sections. $n = 6$ biological replicates for each group. Scale bar, 50 μm. **d** The mRNA levels of KIM1 and neu-trophil gelatinase-associated lipocalin (NGAL) in mice. $n = 3$ biological replicates for each group. **e** Representative micrographs and quantification of TUNEL staining (green) in each group (the line box indicates the magnified image). Scale bar, 50 μm, $n = 6$ biological replicates for each group. **f** Representative images of western blot and quantitative analyses of BAX and Bcl-2. ACTIN was used as the loading control. $n = 5$ biological replicates for each group. The data are presented as the means ± SEMs. All statistical analysis were performed by two-way ANOVA with Tukey's multiple comparison test. Source data are provided as a Source data file.

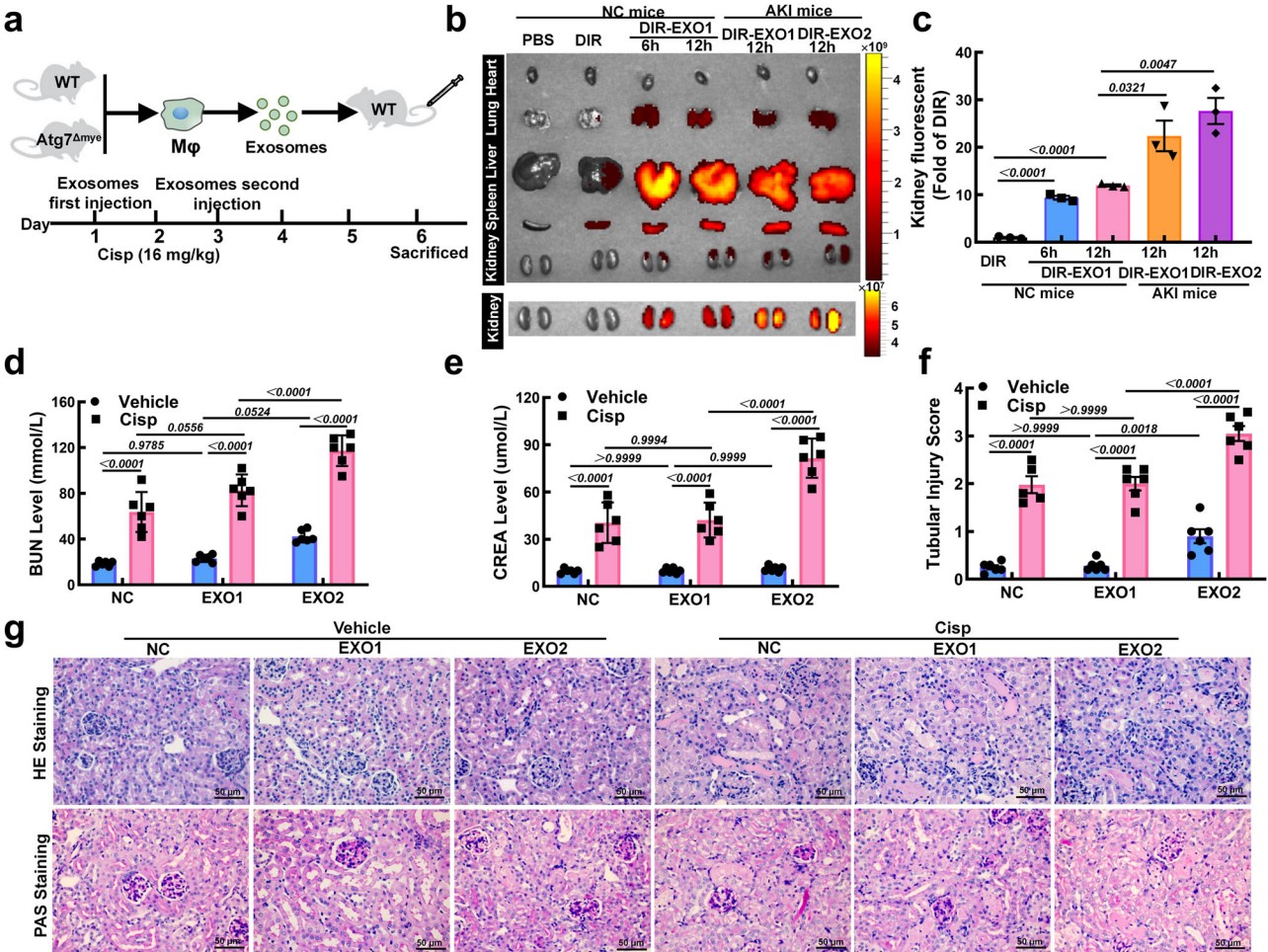

**Fig. 4 | Exosomes derived from ATG7-deficient macrophages (Mφs) exacerbate kidney injury in vivo. a** Experimental procedure to explore the effects of Mφ-derived exosomes on cisplatin-induced AKI. Mice were injected with Mφ-derived exosomes from WT (Mφ^WT-EXO, EXO1) or Atg7^Δmye mice (Mφ^Atg7Δmye-EXO, EXO2) (~100 μg (at the protein level) in 100 μl) 24 h before cisplatin injection (16 mg/kg). **b** Imaging of DiR-labeled exosomes from different tissues. **c** Quantification of the relative fluorescence intensity of the infiltrated exosomes in the kidney in different groups. *n* = 3 biological replicates for each group, unpaired two-tailed Student's t test. **d, e** The serum levels of BUN (**d**) and CREA (**e**) in the mice. *n* = 6 biological replicates for each group. **f, g** Representative images of hematoxylin-eosin (HE)- and periodic acid-Schiff (PAS)-stained kidney sections and the tubular injury scores of the mice. Scale bars, 50 μm. *n* = 6 biological replicates for each group. The data are the means ± SEMs. Statistical analysis were performed by two-way ANOVA with Tukey's multiple comparison test in (**d–f**). EXO1, Mφ^WT-EXO; EXO2, Mφ^Atg7Δmye-EXO. Source data are provided as a Source data file.

prediction. In TECs, the miR-195a-5p mimic significantly inhibited the luciferase activity of SIRT3-WT but not that of SIRT3-MUT or the protein level of SIRT3 (Fig. 7b, c). Moreover, the protective effect of the miR-195a-5p inhibitor on mitochondrial dysfunction was blunted in the presence of SIRT3 knockdown (Fig. 7d–f). To test whether the inhibition of SIRT3 mediates the detrimental effects of miR-195a-5p on kidney mitochondrial function in vivo. Lentivirus packaged SIRT3 (Over-SIRT3) was intraperitoneally injected into the kidney 3 d before cisplatin induction, resulting in approximately 2.51 ± 0.13-fold over-expression in AKI mice (Fig. S7 and Fig. 7g). After the over-SIRT3 treatment, the reduced protein levels of mitochondrial ETC complexes in the cisplatin-treated Atg7^Δmye mice increased (Fig. S7). Accordingly, the destruction of mitochondrial morphology was improved in the SIRT3-overexpressing treated mice (Fig. 7g). Thus, we speculate that autophagy deficiency in Mφ aggravates cisplatin-induced mitochondrial dysfunction via the miR-195a-5p/SIRT3 axis.

### Overexpression of SIRT3 rescued kidney injury from cisplatin in Atg7^Δmye mice

Next, we investigated the protective role of SIRT3 in cisplatin-treated Atg7^Δmye mice. In contrast with the findings in the cisplatin-induced

Atg7^Δmye mice, the serum BUN and CREA levels were decreased in the SIRT3-overexpressing treated mice (Fig. 7h). Kidney morphological abnormalities, tubular damage scores, and KIM1 expression were improved in the SIRT3-overexpressing group (Fig. 7i). Furthermore, the apoptosis caused by cisplatin was alleviated after SIRT3 over-expression, as demonstrated by a reduced number of TUNEL-positive cells and decreased expression of the BAX protein (Fig. S8). Collectively, these results indicate that SIRT3 protects against cisplatin-induced kidney injury in Atg7^Δmye mice.

### Adoptive transfer of autophagy-activated Mφ alleviated kidney injury in AKI mice

To elucidate the specific role of autophagy in Mφ-mediated kidney injury, Mφ from mice with cisplatin-induced AKI were treated with trehalose (a natural nonreducing disaccharide), a novel autophagy inducer[24]. Here, we found that trehalose treatment increased the expression of LC3 II, which was further elevated after HCQ incubation, indicating that autophagy was activated (Fig. 8a). Interestingly, along with the activation of autophagy, miR-195a-5p in Mφ from AKI mice (Mφ^AKI) was reduced (Fig. 8b). Mφ from AKI mice (Mφ^AKI) or trehalose-treated Mφ^AKI mice (Mφ^AKI+Tre) were adoptively transferred into AKI

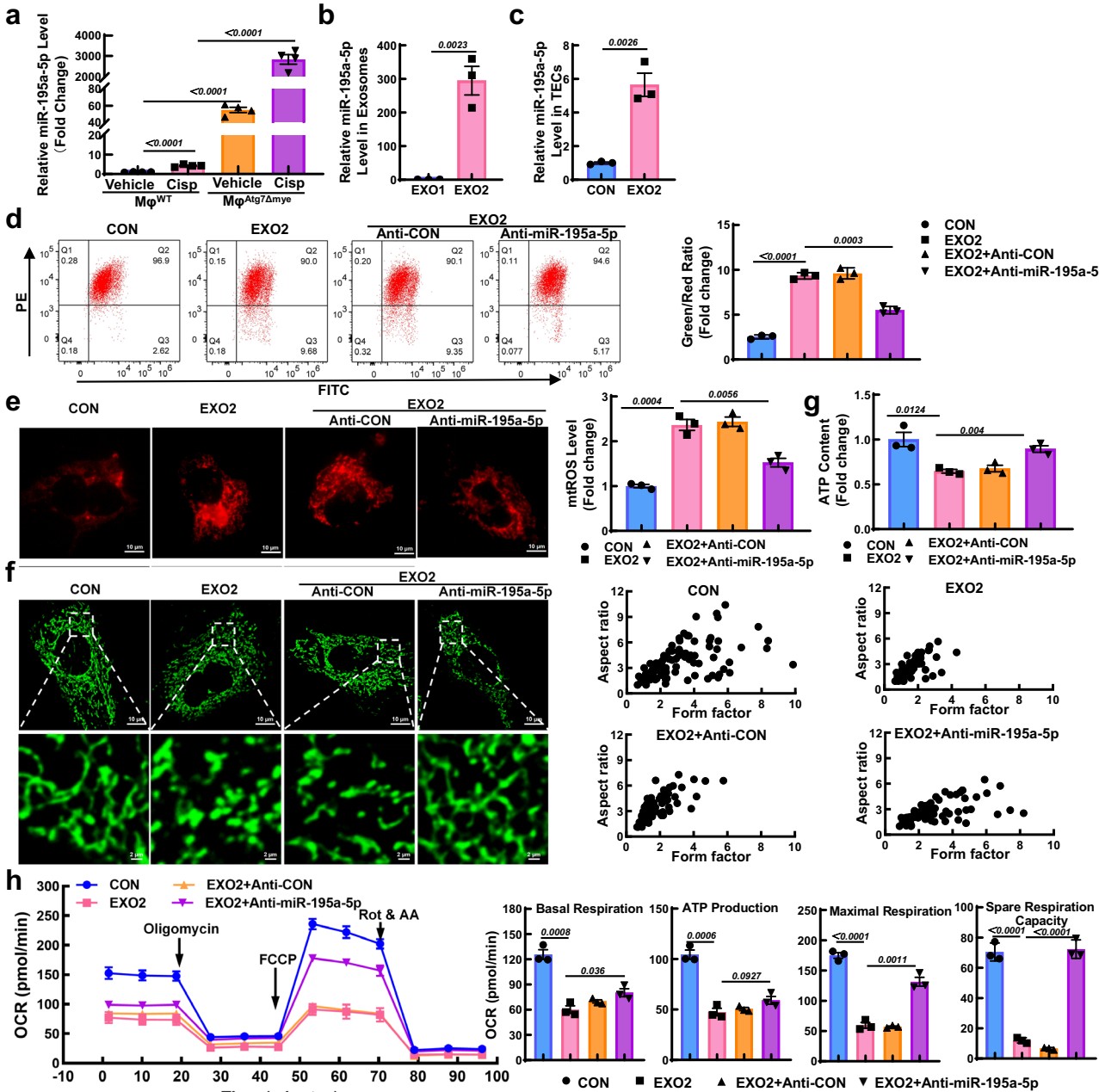

**Fig. 5 | Macrophage (Mφ)-derived miR-195a-5p impairs mitochondria in TECs.**
**a** The level of miR-195a-5p in Mφ from WT and Atg7^Δmye mice. *n* = 4 biological
replicates for each group. **b** The level of miR-195a-5p in Mφ-derived exosomes.
**c** The level of miR-195a-5p in TECs after the incubation with exosomes (10 μg/ml).
*n* = 3 biological replicates for each group. **d** Flow cytometry analysis of the mito-
chondrial membrane potential (Δψm) in TECs. *n* = 3 biological replicates for each
group. **e** Representative images and quantification of mitochondrial ROS (mtROS)
(red) in TECs. Scale bars, 10 μm, *n* = 3 biological replicates for each group
**f** Representative immunofluorescence images and quantification of mitochondrial

morphology (green) in TECs loaded with MitoTracker Green. Scale bars, 10 μm and
2 μm. **g** The ATP content of TECs was measured by an ATP assay kit, and the ATP
concentration was calculated in nmol/mg protein. *n* = 3 biological replicates for
each group. **h** Measurement of the mitochondrial OCR in TECs using a Mito Stress
kit, and the quantification of basal respiration, ATP production, maximal respira-
tion, and spare respiration capacity. *n* = 3 biological replicates for each group. The
data are presented as the means ± SEMs. All statistical analysis were performed by
unpaired two-tailed Student's t test. EXO1, Mφ^WT-EXO; EXO2, Mφ^Atg7Δmye-EXO.
Source data are provided as a Source data file.

mice, where the endogenous Mφ in the mice were first depleted using
Lipo-Clod. After Lipo-Clod injection, renal-infiltrated Mφ were sig-
nificantly depleted, as described above, resulting in the partial inhibi-
tion of mitochondrial morphology destruction and reduced
expression of ETC complexes (Fig. 8c, d). As a consequence of the
improvement in mitochondrial function, kidney injury and apoptosis
in AKI mice were slightly alleviated after Lipo-Clod injection (Fig. 8d–f
and Fig. S9). Moreover, under Mφ depletion conditions, the kidney
injury in AKI mice was aggravated when Mφ^AKI was adoptively

transferred into mice, whereas kidney injury was alleviated after the
adoptive transfer of Mφ^AKI+Tre (Fig. 8d–f and Fig. S9). Collectively,
these data suggest that the activation of autophagy in Mφ can ame-
liorate mitochondrial dysfunction and kidney injury in AKI mice.

## Discussion
As an important component of the innate immune system, Mφ are
involved in defending cells by clearing cellular debris and invading
pathogens and regulating inflammatory responses[29]. Based on the key

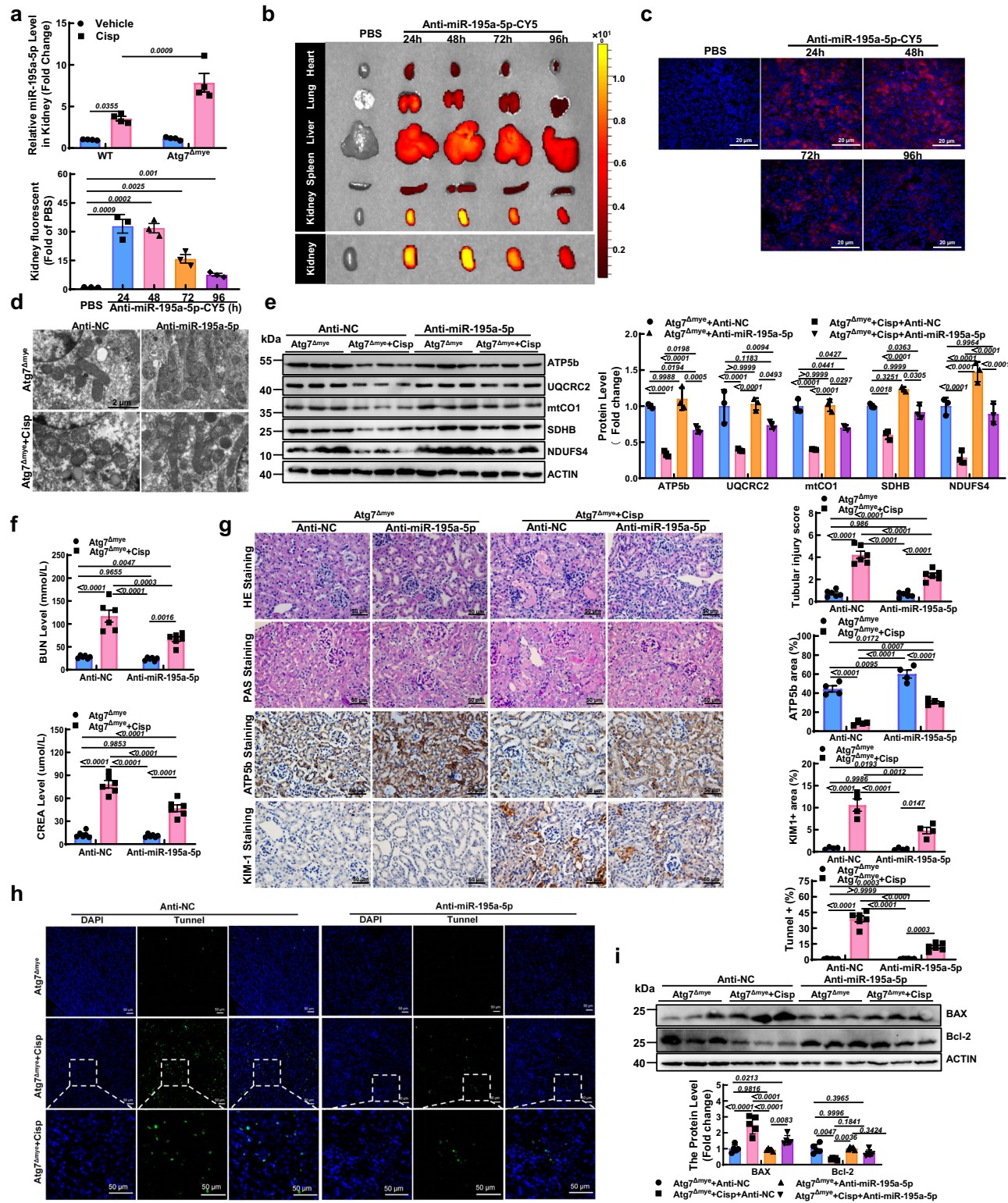

role of autophagy in Mφ polarization[7], we first examined the changes in Mφ autophagy in AKI mice, and found that autophagy was activated first and then inhibited. Consistent with previous findings in obese mice and acute liver injury mice, we confirmed that deficient Mφ autophagy exacerbated cisplatin-induced systemic inflammation.

Recently, exosome-mediated crosstalk between TECs and Mφ has been shown to play a critical role in the development of AKI[15]. The biogenesis and content of exosomes not only are determined by their cellular source but are also sensitive to their cellular status[30]. It has

been reported that more Mφ-derived exosomes are secreted after high-glucose treatment[31]. Similar results were also found in nicotine-treated Mφ[32]. Interestingly, no significant change in the number of exosomes was observed between Mφ from the Atg7$^{\Delta mye}$ mice (EXO2) and those from the WT mice (EXO1). However, the contents in EXO2 were slightly greater than those in EXO1, which may be due to the different stimuli used. Most exosome functions have been attributed to miRNAs that regulate target mRNA expression by binding to specific sequences in the 3′UTR. Exosomal miR-155 derived from adipose tissue

**Fig. 6 | The miR-195a-5p antagomir attenuates mitochondrial dysfunction and kidney injury in cisplatin-induced Atg7$^{\Delta mye}$ mice. a** The level of miR-195a-5p in kidney sections from WT and Atg7$^{\Delta mye}$ mice. $n = 4$ biological replicates for each group. **b** Representative IVIS images of different organs harvested from mice after intravenous injection of anti-miR-195a-5p-5'Cy5 (PBS *vs.* Anti-miR-195a-5p-CY5-24h, $P = 0.0009$; PBS *vs.* Anti-miR-195a-5p-CY5-48h, $P = 0.0002$; PBS *vs.* Anti-miR-195a-5p-CY5-72h, $P = 0.0025$; PBS *vs.* Anti-miR-195a-5p-CY5-96h, $P = 0.001$). $n = 3$ biological replicates for each group, unpaired two-tailed Student's t test. **c** Representative micrographs of anti-miR-195a-5p-5'Cy5 (red) in the kidney. Scale bar, 20 μm. **d** Representative TEM images of kidney tissues from mice. Scale bar, 2 μm. **e** Representative images of western blot and quantitative analyses of OXPHOS-related genes (ATP5b, UQCRC2, mtCO1, SDHB, and NDUFS4). ACTIN was used as the loading control. $n = 3$ biological replicates for each group. **f** The serum levels of BUN and CREA in the Atg7$^{\Delta mye}$ mice. $n = 6$ biological replicates for each group. **g** Representative images of hematoxylin-eosin (HE) and periodic acid-Schiff (PAS)-staining ($n = 6$ biological replicates for each group), immunohistochemical staining and quantification of ATP5b and KIM1 ($n = 4$ biological replicates for each group) in paraffin-embedded kidney sections. Scale bars, 50 μm. **h** Representative micrographs and quantification of TUNEL staining (green) in Atg7$^{\Delta mye}$ mice treated with anti-NC or anti-miR-195a-5p treatment. Scale bars, 50 μm, $n = 6$ biological replicates for each group. **i** Representative images of western blots and quantitative analyses of BAX and Bcl-2 expression. ACTIN was used as the loading control. $n = 5$ biological replicates for each group. The data are the means ± SEMs. Statistical analysis was performed by two-way ANOVA with Tukey's multiple comparison test in (**a**) and (**e**–**i**). Source data are provided as a Source data file.

Mφ (ATMs) in obese mice caused glucose intolerance and insulin resistance when administered to lean mice[33]. Moreover, exosomes released by Mφ in diopathic pulmonary fibrosis (IPF) patients fight pulmonary fibrosis progression via delivery of antifibrotic miR-142-3p to alveolar epithelial cells and lung fibroblasts[34], indicating that miRNA expression in Mφ changes under different conditions, which may limit or promote the injury response. Herein, the miR-195a-5p levels in both Mφ- and Mφ-derived exosomes from the Atg7$^{\Delta mye}$ mice were significantly higher than those from the WT mice. In addition, along with the uptake of Mφ-derived exosomes, the miR-195a-5p level in TECs was also elevated. miR-195a-5p, a member of the micro-15/16/195/424/497 family, is stress-inducible, and can be activated in multiple diseases, such as cancers[35], schizophrenia[36], sepsis[37], and heart failure[38,39]. Interestingly, previous studies have reported that miR-195a-5p promotes apoptosis and mitochondrial dysfunction[39,40]. Similar to the findings of these previous studies, we found that the mitochondrial function of the kidney in Atg7$^{\Delta mye}$ mice and EXO2-treated TECs was impaired, but improved with the miR-195a-5p antagomir.

Our previous study confirmed that reduced SIRT3 in AKI mice led to hyperacetylation of OPA1 and ATP5b synthase which are associated with mitochondrial dynamics and impaired mitochondrial respiration[41]. Given the renoprotective effect of SIRT3 on mitochondrial dysfunction in AKI mice[42,43], as well as the direct miR-195a-5p-mediated suppression of SIRT3 in the heart, we hypothesized that cisplatin-induced severe mitochondrial damage in Atg7$^{\Delta mye}$ mice may be due to the miR-195-SIRT3 axis. As expected, the luciferase activity of SIRT3-WT was inhibited by the miR-195a-5p mimic, suggesting that SIRT3 is a direct target mRNA of miR-195a-5p. Moreover, similar to the effects of the miR-195a-5p antagomir, in vivo overexpression of SIRT3 restored mitochondrial function and alleviated kidney injury in Atg7$^{\Delta mye}$-AKI mice.

To date, increasing evidence has confirmed that cisplatin is mainly excreted by the kidney and accumulates in renal TECs through organic cation transporters, indicating that TECs injury may be the initial event in the process of cisplatin-induced AKI[44,45]. Moreover, several studies have shown that exosomes from injured TECs in AKI mice can activate proinflammatory Mφ and induce Mφ infiltration in renal tissue[19,46]. Thus, we deduced that the multidirectional intercellular communication between TECs and Mφ may be involved in the process of AKI.

However, there are several limitations. First, ATG7 can exert autophagy-independent effects. Yao et al. demonstrated that ablation of endothelial ATG7 could inhibit ischemia-induced angiogenesis by upregulating Stat1 to suppress Hif1α expression[47]. Moreover, the absence of ATG7 resulted in increased DNA damage and increased p53-dependent apoptosis during metabolic stress[48]. Although our study demonstrated that impaired Mφ autophagy is crucial for aggravated kidney injury in Atg7$^{\Delta mye}$ mice, the ATG7-mediated autophagy-independent mechanism may contribute to cisplatin-induced kidney injury in AKI mice[49]. Second, the Lyz2 promoter is known to confer efficient activity not only in monocytes/Mφ but also in other cells of myeloid origin, such as neutrophils and eosinophils, under steady-state conditions[50]. Although Mφ depletion significantly protected against cisplatin-induced kidney injury in Atg7$^{\Delta mye}$ mice, further investigation of whether the ATG7 in this cell subset contributing to kidney injury in Atg7$^{\Delta mye}$ mice is necessary. Finally, it would be more convincing if our experiments were verified in different AKI models, such as an ischemia-reperfusion model.

In summary, we have demonstrated that autophagy deficiency in Mφ induces systemic inflammation, impairs mitochondria, and aggravates kidney injury in AKI mice. Moreover, autophagy deficiency in Mφ impairs mitochondria in TECs via exosomal miR-195a-5p transfer. Finally, targeting the miR-195a-5p/SIRT3 axis alleviates cisplatin-induced kidney injury in the Atg7$^{\Delta mye}$ mice (Fig. 8G). These findings indicate that Mφ autophagy may be an important therapeutic target in AKI and other inflammation-related diseases.

## Methods

### Ethics statement
All animal experiments described in this study were approved by the Animal Care and Use Committee of West China Hospital, Sichuan University (No. 2018193A), and were conducted according to the National Institutes of Health Guide for the Care and Use of Laboratory Animals.

### Study design
The objectives of this study were to examine the role of Mφ autophagy in the pathogenesis of AKI and the underlying mechanisms involved. We first designed a cisplatin-induced AKI model to determine changes in Mφ autophagy. To investigate the in vivo role of Mφ autophagy in AKI mice, Atg7$^{f/f}$ mice[46] with floxed alleles for the autophagy gene ATG7 were crossed with Lyz2-Cre mice with the mouse lysozyme M promoter-driven Cre recombinase to generate myeloid cell-specific deletion of ATG7 mice (Atg7$^{\Delta mye}$). After cisplatin injection, renal function, histology, and systemic inflammation were analyzed. To understand whether exosome delivery-mediated crosstalk between Mφ and TECs affects kidney injury, Mφ-derived exosomes from WT and Atg7$^{\Delta mye}$ mice were isolated and then injected into mice via the tail vein in vivo. To study the molecular mechanisms underlying the aggravated kidney damage induced by autophagy deficiency in Mφ, AKI mice were treated with the miR-195a-5p antagomir and SIRT3 overexpression lentivirus. For animal studies, animals were chosen at random for the vehicle or treatment groups. An experimental group size of six or more animals was used to detect at least a 20% difference with a power of 95% and $P < 0.05$. For in vitro experiments, primary TECs stimulated with Mφ-derived exosomes were treated with a miR-195a-5p inhibitor, and mitochondrial morphology and mitochondrial bioenergetics were determined. A minimum of three experimental replicates were performed. The numbers of replicates are presented in the figure legends.

### Animals
C57BL/6 mice (male, 8–10 weeks old) were obtained from Chengdu Dashuo Laboratory Animal Technology Co. Atg7$^{f/f}$ mice[46] (Stock NO.

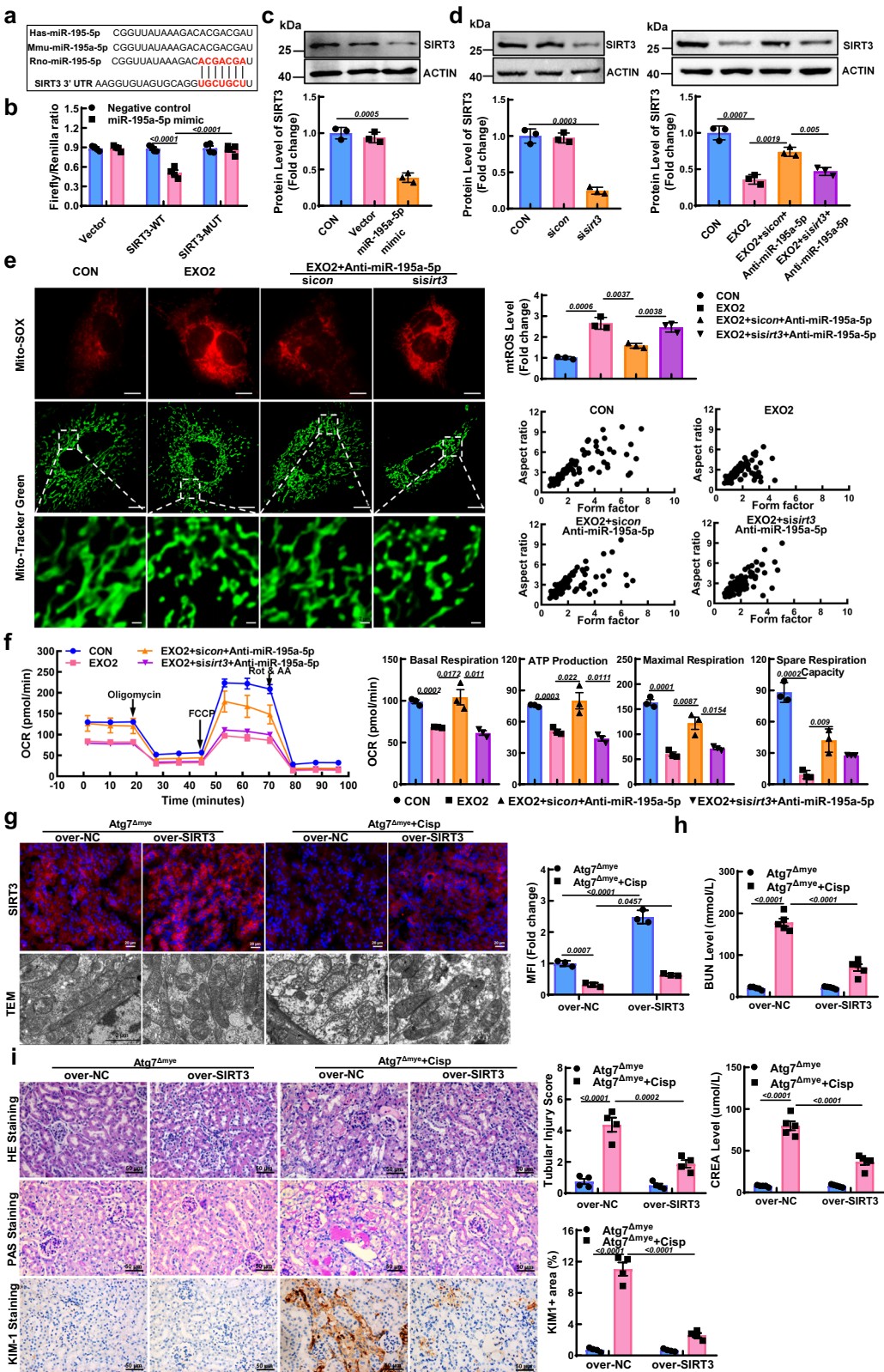

RBRC02759; Tsukuba, Japan) were generated by insertion of loxP site within introns 13 and 14. Exon 14 was fused to a cDNA fragment encoded by exons 15, 16, and 17 and polyA after the stop codon. The details of their genetic background are as follows: B6.Cg-ATG7 <tm1Tchi>. To generate myeloid cell-specific deletion of ATG7 mice (Atg7$^{\Delta mye}$), Atg7$^{f/f}$ mice were crossed with Lyz2-Cre mice (Stock NO.004781; Shanghai Nan Fang Model Organism Research Center,

Shanghai, China) with the mouse lysozyme M promoter-driven Cre recombinase. The control mice for all experiments were littermate Atg7$^{f/f}$ mice lacking the Cre transgene. All mice housed in the Animal Center of West China Hospital, Sichuan University under standard conditions with free access to food (1010082; Xietong Pharmaceutical Bio-engineering Co., Ltd., Jiangsu, China) and water. The light was from 7 am to 7 pm, with the temperature kept at 21–24 °C and humidity at

**Fig. 7 | Overexpression of SIRT3 alleviates EXO2-induced mitochondrial damage and cisplatin-induced kidney injury. a** Prediction of SIRT3 as a target of miR-195a-5p. **b** Dual-luciferase assays of TECs cotransfected with a SIRT3 luciferase reporter (pmirGLO-SIRT3-WT, SIRT3-WT) and a miR-195a-5p mimic or mutant SIRT3 luciferase reporter (pmirGLO-SIRT3-MUT, SIRT3-MUT), and the luciferase activity of the cells was detected using a dual-luciferase assay kit. $n = 4$ biological replicates for each group, two-way ANOVA with Tukey's multiple comparison test. **c**, **d** Western blot and quantitative analyses of SIRT3 in TECs treated with the treatment of miR-195a-5p mimic, or miR-195a-5p inhibitor. ACTIN was used as the loading control. $n = 3$ biological replicates for each group. **e** Representative images and quantification of mitochondrial ROS (mtROS) (red) and mitochondrial morphology (green) in TECs. Scale bars, 10 μm and 2 μm, $n = 3$ biological replicates for each group. **f** Measurement of the mitochondrial OCR in TECs using a Mito Stress kit and quantification of basal respiration, ATP production, maximal respiration,

and spare respiration capacity. $n = 3$ biological replicates for each group. Statistical analysis were performed by unpaired two-tailed Student's t test in (**c**–**f**). For lentiviral experiments, cisplatin-induced AKI was induced 3 days after intrarenal injection of lentivirus carrying SIRT3 (LV-*Sirt3*) or NC (LV-nc). **g** Representative micrographs of SIRT3 (red) and TEM images of the kidneys of Atg7$^{Δmye}$ mice. Scale bars, 20 μm and 2 μm, $n = 3$ biological replicates for each group. **h** Serum levels of BUN and CREA in mice. $n = 5$ biological replicates for each group. **i** Representative images of hematoxylin-eosin (HE) and periodic acid-Schiff (PAS)-staining, and immunohistochemical staining and quantification of KIM1 expression in kidney sections. Scale bars, 50 μm, $n = 4$ biological replicates for each group. Statistical analysis was performed by two-way ANOVA with Tukey's multiple comparison test in (**g**–**i**). The data are the means ± SEMs. Source data are provided as a Source data file.

40–70%. Male mice were randomly divided into different groups as specified. All experimental procedures involving animals were in accordance with the guidelines of the Animal Care and Use Committee of West China Hospital, Sichuan University (Chengdu, China).

### Mouse AKI model

To generate the cisplatin-induced AKI model, cisplatin (16 mg/kg, MCE, Shanghai, China) was administered by a single intraperitoneal injection. For the normal control (NC), the mice received an intraperitoneal injection of an equal volume of 0.9% saline. Four days after cisplatin injection, mice were anesthetized with sodium pentobarbital, after which the mice were sacrificed to collect blood samples for serum creatinine and BUN measurements, and renal tissues were collected and stored at −80 °C, or formalin-fixed for subsequent histological analysis.

### Biochemistry assay

Blood samples were separated by centrifugation at $1000 \times g$ for 15 min, and the renal function parameters (BUN and CREA) in the serum were measured by the Department of Laboratory Medicine of West China Hospital (Chengdu, China).

### Histology and immunohistochemistry

Kidney histology was determined on formalin-fixed sections stained with hematoxylin-eosin and periodic acid-Schiff. The tissue injury score was graded by cell necrosis, loss of brush border, cast formation, and tubular degeneration (0: none; 1: <10%; 2: 11–25%; 3: 26–45%; 4: 46–75%; and 5: >76%) as previously reported[51]. For immunohistochemical staining, after quenching with endogenous peroxidase, antigen retrieval, and blocking nonspecific binding sites, the kidney tissue sections were incubated with anti-KIM1 (AF1817; RD Systems, Canada), anti-TNF-α (ab215188; Abcam, Cambridge, UK), and anti-ATP5b (A5769; ABlonal; Wuhan, China) overnight at 4 °C, followed by staining with horseradish peroxidase (HRP)-conjugated secondary antibodies (ABclonal) and 3,3'-diaminobenzidine (DAB) substrates. Images of stained sections were captured by a light microscope (Zeiss, AX10 imager A2, Oberkochen, Germany) and quantified by ImageJ software.

### Enzyme-linked immunosorbent assay (ELISA)

The levels of IL1β and TNF-α in the serum were determined using commercial ELISA kits (Thermo Fisher Scientific, USA) following the recommended protocols.

### Flow cytometry

Kidneys were harvested and cut into small fragments, and the tissue pieces were collected and dissociated with 1 mg/mL collagenase I (Gibco, USA) for 30 min at 37 °C. Then, the digestion solutions were filtered through 100 μm and 40 μm cell strainers (Falcon, USA) to obtain single-cell suspensions. Primary renal Mφ labeled with F4/80+

were sorted from mice with a FACSAria SORP cytometer (BD Biosciences, USA) and cultured with Mφ medium.

To analyze the phenotype of Mφ, flow cytometric identification was performed using combinations of the following mAbs: F4/80-APC (123116; Biolegend, USA), CD206-PE (141706; Biolegend), and CD11c-FITC (N418; Biolegend). After incubation for 30 min, flow cytometry analysis was performed on a CytoFLEX LX flow cytometer (Beckman Coulter, USA).

To detect the expression of LC3 in kidney-infiltrated Mφ (KM), the cells were blocked with Fc Block (BD Bioscience, USA) for 5 min and stained with the following antibodies: F4/80-PE (QA17A29; Biolegend) and LC3B (E5Q2K; CST). The concentration-matched mouse mAb IgG2b (E7Q5L; CST) was used as the isotype control. Alexa Fluor® 594-conjugated goat anti-mouse IgG (Poly4053; Biolegend) was used as a secondary antibody.

### Immunofluorescence

For immunofluorescence, sections were prepared according to standard procedures, and then incubated with anti-F4/80 (ab6640; Abcam) or anti-LC3 (2775; Cell Signaling Technology, MA, USA), followed by incubation with goat anti-rat IgG H&L (DyLight® 550) (ab9688; Abcam). Finally, the sections were treated with DAPI for nuclear staining and detected by confocal microscopy (Nikon, TiA1-N-STORM, Tokyo, Japan).

### Western blot

Renal tissue and cells were lysed in RIPA lysis buffer containing the protease inhibitor PMSF. Protein concentration was determined by bicinchoninic acid (BCA) reagent (Beyotime Biotechnology, Shanghai, China). For western blot analysis, equal amounts of proteins were loaded on SDS–polyacrylamide electrophoresis gels, and then transferred to PVDF membranes (Roche, Switzerland), followed by incubation with the appropriate primary and secondary antibodies (Table S1). The blots were visualized with Immobilon™ Western Chemiluminescent HRP Substrate.

### Real-time PCR (RT–PCR)

Total RNA was extracted from renal tissues with TRIzol, and then reverse-transcribed to cDNA by using a Transcriptor First Stand cDNA Synthesis Kit and random primers. Quantitative RT–PCR was performed using a SYBR Green PCR master mix kit with specific primers (Table S2) on a Chromo4 cycler (Bio-Rad, Hercules, CA, USA). Data analysis was performed by using the ΔΔCt method. The change in mRNA levels in the samples was determined using the delta-delta Ct method with ACTIN as the reference gene.

For miR-195a-5p quantification, RT–PCR was performed using SYBR Green Master Mix with the miR-195a-5p-specific forward primer and the universal reverse primer (RiboBio, Guangzhou, China). U6 snRNA was used as an internal control to normalize the expression of miR-195a-5p.

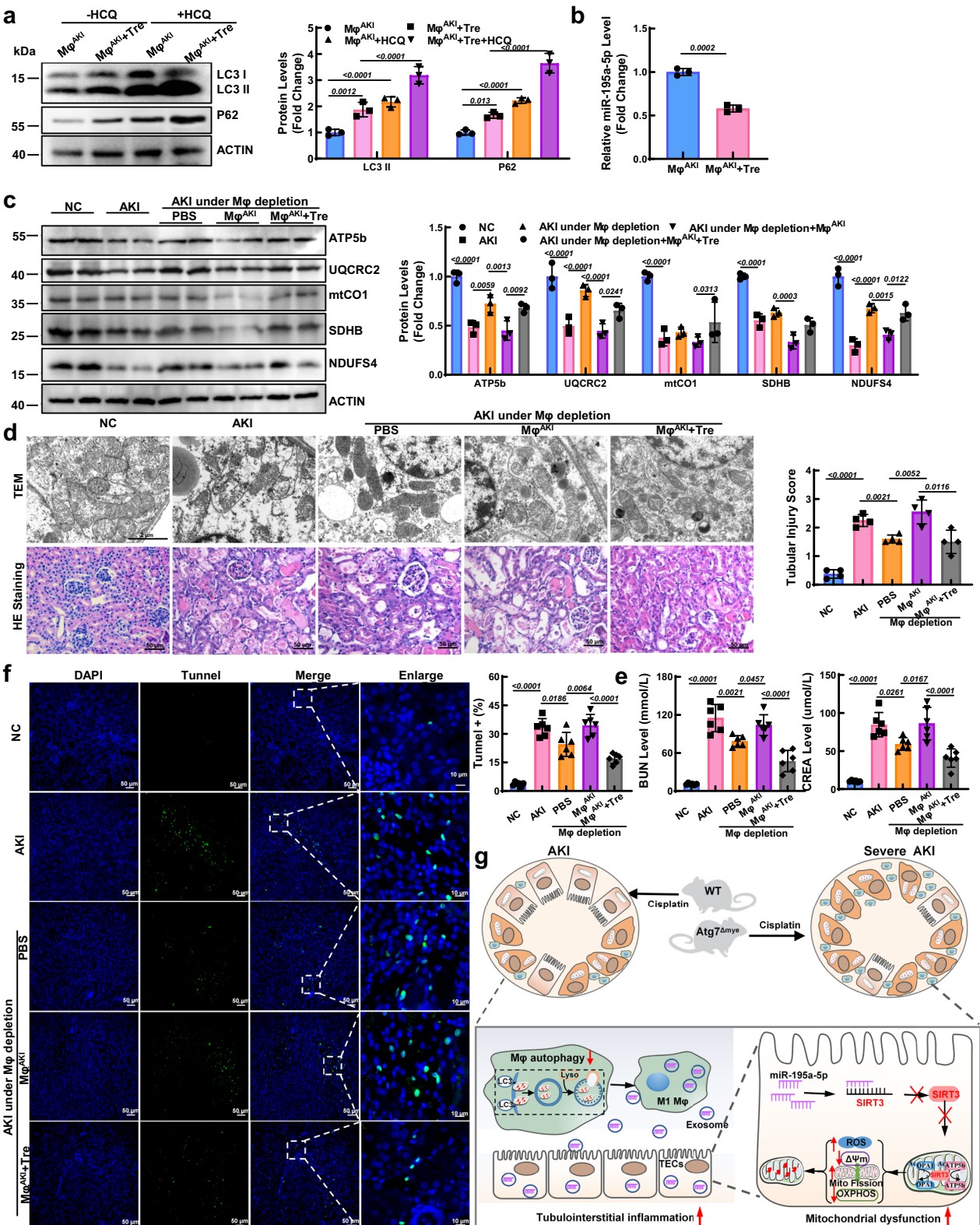

**Terminal deoxynucleotidyl transferase-mediated dUTP nick-end labeling (TUNEL) assay**

Paraffin-embedded slides were analyzed for DNA strand breaks by fluorescent enzymatic labeling of the free 3'OH termini of modified nucleotides using a commercially available kit (In Situ Apoptosis Detection Kit; Merck Millipore). The slides were visualized using a confocal microscope set at excitation/emission wavelengths of 488/510 nm.

**In vitro and in vivo anti-miR-195a-5p treatment**

For in vitro treatment, tubular epithelial cells (TECs) treated with EXOs were transfected with miR-195a-5p inhibitor (200 nM) or inhibitor negative control (200 nM) (RiboBio Co., Ltd.) using a FECT CP Transfection Kit (RiboBio) according to the manufacturer's protocols.

For in vivo treatment, oligonucleotides targeting miR-195a-5p (a miR-195a-5p antagomir) or a control antagomir were obtained from by

**Fig. 8 | Adoptive transfer of autophagy-activated macrophages (Mφ) alleviates kidney injury in AKI mice. a** Peritoneal Mφ isolated from AKI mice were treated with trehalose, after which the protein levels of LC3 II and P62 were measured. ACTIN was used as the loading control. $n = 3$ biological replicates for each group, two-way ANOVA with Tukey's multiple comparison test. **b** The level of miR-195a-5p in Mφ from AKI mice treated with trehalose. ACTIN was used as the loading control. $n = 3$ biological replicates for each group, unpaired two-tailed Student's t test. After endogenous Mφ were deleted by clodronate liposomes (Lipo-Clod), AKI was induced in mice by cisplatin. Then, Mφ from AKI mice (Mφ$^{AKI}$) or trehalose-treated Mφ$^{AKI}$ mice (Mφ$^{AKI}$+Tre) were adoptively transferred into the mice at 6 h after cisplatin treatment. **c** western blot and quantitative analyses of OXPHOS-related proteins. ACTIN was used as the loading control. $n = 3$ biological replicates for each group, two-way ANOVA with Tukey's multiple comparison test. **d** Representative TEM images and hematoxylin-eosin (HE) staining of kidneys from AKI mice after adoptive transfer of Mφ. Scale bars, 2 μm and 50 μm, $n = 4$ biological replicates for each group, unpaired two-tailed Student's t test. **e** Renal function (serum levels of BUN and CREA) detection. $n = 6$ biological replicates for each group, one-way ANOVA with Tukey's multiple comparisons test. **f** Representative micrographs and quantification of TUNEL staining (green) in each group. Scale bars, 20 μm and 50 μm, $n = 6$ biological replicates for each group, one-way ANOVA with Tukey's multiple comparisons test. **g** Schematic representation of the data. Data in (**a**) and (**b**) show a representative of three independent experiments. The data are the means ± SEMs. Source data are provided as a Source data file.

GenePharma (Shanghai, China). To deliver the antagomir, mice were administered 50 nmol of the antagomir via the tail vein for 24 h and then administered cisplatin (16 mg/kg, i.p.).

## Intrarenal lentivirus delivery

The SIRT3 overexpression lentivirus (LV-*Sirt3*), generated by subcloning sirt3 (NM_022433.2) into the shuttle vector LV5 (EF-1a/GFP&Puro), was obtained from GenePharma (Shanghai, China). LV-*Sirt3* was administered via renal cortex injection according to previous reports[52,53]. Briefly, mice were anesthetized with pentobarbital sodium (30 mg) by intraperitoneal injection and then injected with $1 \times 10^9$ TU/ml lentivirus (LV-Sirt3 or LV-nc) at 5 different sites (10 μl at each site) 3 days before cisplatin injection.

## Isolation of peritoneal Mφ (PM)

After the mice were anesthetized, 5–8 ml of saline was injected into the peritoneal cavity. Then, the injected fluid was collected by centrifugation at $500 \times g$ for 5 min. The cell pellets were resuspended in RPMI-1640 media (R8758; Sigma-Aldrich) supplemented with 10% FBS. After the cells were allowed to adhere overnight, the culture medium was changed, and the cells were used for further experiments. To induce autophagy, primary Mφ were treated with trehalose (100 mM) for 48 h.

## Adoptive transfer of Mφ

Endogenous Mφ were depleted using clodronate liposomes (Lipo-Clod; FormuMax, Scientific, Palo Alto, CA) (200 μl, i.p.). After the Mφ were removed on Day 3, AKI was induced in the mice by cisplatin as described herein. Mφ from AKI mice (Mφ$^{AKI}$) or trehalose-treated Mφ$^{AKI}$ (Mφ$^{AKI}$+Tre) ($5 \times 10^5$ cells/mouse) were adoptively transferred into the mice at 6 h after cisplatin treatment.

## Exosome (EXO) purification and characterization

The EXOs from primary Mφ culture medium were prepared as previously described[54]. Briefly, all samples were centrifuged at $300 \times g$ for 15 min, followed by centrifugation at $2000 \times g$ for 15 min to eliminate dead cells and debris. Then, the supernatant was centrifuged at $10,000 \times g$ for 30 min and filtered through a 0.2 mm filter (Merck Millipore; Billerica, MA, USA). After that, the supernatant was ultracentrifuged at $130,000 \times g$ for 2 h to pelletize the EXOs using a SW41 rotor (Beckman Coulter). After washing with cold PBS at the same speed, the EXOs were resuspended in 100 μl of 0.22 μm filtered cold PBS. To characterize the EXOs, the particle size and concentration were measured by Zeta View. Total protein expression in exosomes collected from $1 \times 10^6$ cells was measured by a MicroBCA Protein Assay Kit (Thermo Fisher Scientific).

## Treatment of TECs with EXOs

TECs were seeded into six wells, and then 10 μg/ml EXOs secreted from Mφ were added. After being incubated for 48 h at 37 °C, the cells were harvested for cell survival assays.

## Dual-luciferase reporter assay

The plasmids were all obtained from Shanghai Genechem Co., Ltd. TECs were transfected with a dual-luciferase reporter plasmid containing the full sequence (wild-type) of the SIRT3 3′UTR (pmir-GLO-SIRT3-WT) or a mutant sequence of the SIRT3 3′UTR (pmir-GLO-SIRT3-MUT) and a miR-195a-5p mimic using Lipofectamine 3000 according to the manufacturer's instructions (Invitrogen; Thermo Fisher Scientific, Inc.). Following transfection for 6 h at 37 °C, the medium was removed, and fresh medium was added. After 48 h of transfection, the relative luciferase activity of the cells was detected using a dual-luciferase assay kit (cat. no. E1910; Promega Corporation).

## Exosome labeling and cellular uptake

To monitor EXO trafficking, purified EXOs were labeled with the membrane-labeling dye PKH67 (Invitrogen, USA) according to the manufacturer's instructions and were then washed and resuspended in serum-free medium. Then, the PKH67-labeled EXOs were cocultured with CM-Dil-labeled cells for 12 h. Afterward, the cells were fixed with 4% PFA, and the nuclei were labeled with DAPI (Sigma, USA) at 37 °C for 5 min. Then, the stained cells were washed and observed under a microscope (Zeiss, Imager Z2, Oberkochen, Germany).

## In vivo biodistribution of exosomes in mice

Exosomes (~1 μg/μl at the protein level) were incubated with DiR (Invitrogen) at a volume ratio of 500:1 for 30 min, after which the free dye was removed by another round of exosome isolation. After that, 100 μg of DiR-labeled exosomes or PBS was injected into NC or AKI mice via the tail vein, and the biodistribution of exosomes in the whole body and individual organs was detected by an optical imaging system (IVIS Spectrum, PerkinElmer, Waltham, MA, USA). In addition, frozen renal tissue sections from mice were stained with DAPI for 10 min and observed under a fluorescence microscope.

## Transmission electron microscopy (TEM)

For exosome ultrastructural observation, purified EXOs were loaded on carbon-coated copper electron microscopy grids for 2 min. Then, the samples were negatively labeled with 2% phosphotungstic acid for 5 min. Afterward, the grids were washed three times with PBS to remove the additional phosphotungstic acid solution, and the grids were subsequently air-dried and observed via TEM (H-600, Hitachi, Ltd., Tokyo, Japan) at a voltage of 75 kV.

Kidney tissues were fixed in paraformaldehyde and glutaraldehyde and postfixed in osmium tetroxide (201030; Sigma-Aldrich, Taufkirchen, Germany), followed by ethanol dehydration and resin embedding. After that, the tissues were cut into ultrathin sections and stained with uranyl acetate, followed by staining with lead citrate (Sigma-Aldrich, Taufkirchen, Germany). The stained tissue sections were observed under an FEI Tecnai T20 transmission electron microscope.

## Primary TEC isolation and treatment

Mouse primary tubule epithelial cells (TECs) were isolated via a modified protocol as described previously[24]. Briefly, kidney tissues were dissected visually in ice-cold phosphate-buffered saline (PBS) and sliced into ~1 mm³ pieces. The fragments were transferred to a collagenase solution (1 mg/ml, Sigma-Aldrich) and digested at 37 °C for 30 min. Afterward, the supernatant was further sieved through two nylon sieves (pore sizes 100 μm and 40 μm). The proximal tubule (PT) fragments remaining in the 40 μm sieve were resuspended by flushing the sieve in the reverse direction with warm media and then centrifuged at $170 \times g$ for 5 min, washed, and resuspended in DMEM/F-12 medium supplemented with 10% FBS. Finally, the cells were maintained at 37 °C in 5% $CO_2$ in a humidified incubator. For treatment, TECs were cultured in exosome-depleted medium supplemented with or without EXOs (10 μg/ml), miR-195a-5p inhibitors (200 nM) or mimics (20 nM).

## Cell viability and apoptosis assay

Cell viability was determined by a cell counting kit 8 (CCK-8) assay (Dojindo, Japan) and quantified by an enzyme-linked immunosorbent assay (ELISA) microplate reader (BioTek) at 450 nm. For apoptosis determination, the collected cells were incubated with Annexin V–FITC and propidium iodide (PI) for 30 min in the dark and then analyzed via a flow cytometer. According to Nature Protocols[55], living Annexin V-PI- and Annexin V + PI- cells are considered apoptotic cells.

## Small interfering RNA (siRNA) and cell transfection

The sequences of the SIRT3 siRNAs (sisirt3) used were as follows: sense, 5′-GACCUUUGUAACAGCUACATT-3′; antisense, 5′-UGUAGCU-GUUACAAAGGUCTT-3′. TECs at passage three were transfected with sisirt3 and sicon (GenePharma, Shanghai, China) by using jetPRIME® transfection reagent (Polyplus, Illkirch, France) according to the manufacturer's instructions.

## Mitochondrial bioenergetic assay

To measure the mtROS level and mitochondrial membrane potential ($\Delta\psi m$), the cells were incubated with MitoSOX Red (2.5 μM, Thermo Fisher Scientific, Sunnyvale, CA, USA) or JC-1 (5 nM, AAT Bioquest, Sunnyvale, CA, USA) for 15 min at 37 °C. After incubation, the mtROS level was assessed via confocal microscopy, and the $\Delta\psi m$ was analyzed by flow cytometry.

For mitochondrial morphology analyses, cells were stained with MitoTracker Green at 37 °C for 30 min and then visualized with a laser scanning confocal microscope. Mitochondrial length and complexity were reflected by the aspect ratio (AR, major axis/minor axis) and form factor (FF, $4\pi \times$ (area/perimeter²)), respectively, which were calculated with ImageJ software (Wayne, USA).

Cellular ATP levels were determined using a commercially available ATP measurement kit (Beyotime, China) according to the manufacturer's protocol, and the ATP level is presented as nanomoles per milligram of protein.

Mitochondrial OCR analysis was conducted with a Seahorse XF-24 flux analyzer (Seahorse Biosciences, Agilent, USA) using a mitochondrial stress test (MST) assay kit as previously described[56]. Specifically, the concentrations of oligomycin, FCCP, and antimycin A/rotenone were 1, 2, and 0.5 μM, respectively.

## Statistics and reproducibility

To ensure the reproducibility, all the H&E, immunofluorescence and immunohistochemical staining, micrographs of anti-miR-195a-5p-5′ Cy5 and TEM images in the kidney, and western blot were conducted in at least three biological replicates, unless mentioned otherwise. For the analysis of western blot, the same experiment and that gels/blots were processed in parallel. ACTIN was used as the loading control. All data were presented as the means ± SEMs. All the statistical analyses were performed by GraphPad Prism 9. Unpaired two-tailed Student's t test was used for comparisons between 2 groups, one-way ANOVA was used comparisons among multiple groups, and two-way ANOVA was used to determine significance between multiple groups with two experimental parameters. A value of $P < 0.05$ was considered to indicate statistical significance. In the Figure legends, "$n$" indicates the number of biological independent samples.

## Reporting summary

Further information on research design is available in the Nature Portfolio Reporting Summary linked to this article.

## Data availability

All data are available in the main text or the supplementary materials. Source data including all raw data generated in this study are provided with this paper. All data supporting the findings of this study are available from the corresponding authors upon request. Source data are provided with this paper.

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

## Acknowledgements

We thank Yongjie Zhou for assistance with the construction and characterization of Atg7$^{\Delta mye}$ mice. We also thank Qinghua Yin for the evaluation of tissue injury score in animal experimentation. This work was supported by grants from the National Natural Science Foundation of China (82170075 to Y.L.; and 82170590 to Y.C.), The Project of Sichuan Provincial Department of Science and Technology (2022NSFSC1329 to F.L.).

## Author contributions

Y.Y., L.Y., Y.C., and Y.L. designed the study. Y.Y., L.Y., J.Y., F.L., and L.L. performed the experiments. Y.Y., L.Y., S.L., G.L., X.T., J.L., and J.C. analyzed the data. Y.Y., L.Y., Y.C., and Y.L. wrote and revised the manuscript.

## Competing interests

The authors declare no competing interests.
