## [Peer Review File · Nature Communications]

Autophagy-deficient Macrophages Exacerbate
Cisplatin-Induced Mitochondrial Dysfunction and Kidney Injury
via miR-195a-5p-SIRT3 axisREVIEWER COMMENTS

Reviewer #1 (Remarks to the Author):

The manuscript by Yuan et al. reveals that M ϕ autophagy is initially activated and then inhibited in cisplatin-induced AKI mice. The authors demonstrate that the M ϕ -specific deletion of ATG7 exacerbates kidney injury, leading to tubulointerstitial inflammation in cisplatin-induced AKI mice. Importantly, the authors illustrate the involvement of M ϕ -derived exosomes from Atg7 Δ mye mice, particularly miR-195a-5p, in impairing mitochondria in tubular epithelial cells by interacting with SIRT3. Moreover, the authors demonstrate that inhibiting miR-195a-5p and overexpressing SIRT3 enhance mitochondrial function and improve renal function in cisplatin-induced Atg7 Δ mye AKI mice. Finally, the authors show that adoptive transfer of M ϕ from AKI mice to M ϕ -depleted mice exacerbates the kidney injury response to cisplatin. However, this effect is mitigated when M ϕ autophagy is activated with trehalose.

2. Comments to the Authors

The study by Yuan et al. pertains to a crucial area of research. The authors highlight M ϕ autophagy as a crucial contributor to AKI progression and identify miR-195a-5p/SIRT3 axis as a potential therapeutic target. However, there are several major and crucial points that need to be addressed to meet the standards of Nature Communications and engage its readers.

Major Comments

- a. To better understand the role of macrophages in AKI, the authors are encouraged to conduct experiments involving macrophage depletion. This approach will enable them to determine the specific contribution of macrophages, with a particular focus on autophagy, in injury during AKI.
- b. It is essential to measure M1 macrophage markers in addition to CD11c, including CD86, iNOS, Fas ligand, as well as the M2 macrophage markers including CD163 and Arg-1. A panel of markers is often employed to characterize their polarization states accurately.
- c. It is also essential to assess the effect of chemokines on macrophage polarization in AKI.
- d. In the introduction, the authors referenced a study demonstrating that the loss of mitophagy proteins PINK1 and Parkin exacerbated macrophage-derived kidney fibrosis. Therefore, it is highly encouraged for the authors to assess the levels of Parkin and PINK1 proteins when evaluating mitochondrial function. This will help elucidate whether Atg7 Δ mye mice exhibit mitochondrial dysfunction as a result of impaired mitophagy.
- e. In Figure 1c, could the authors please specify if the bar graph represents LC3-II/LC3-I or LC3-II normalized to the loading control actin.

f. The authors mentioned a dramatic decrease in the percentage of M ϕ with LC3-II puncta on the 4th day after cisplatin treatment (Fig. 1d). Could the authors please provide quantification for LC3 staining in peritoneal M ϕ after treatment with hydroxychloroquine (HCQ, 20 μ M) following cisplatin injection for 4 days to justify their statement?

g. In Figure 1d, the authors utilized hydroxychloroquine, an autophagy inhibitor, to assess LC3 puncta. It is crucial to discuss and comment on autophagic flux in the results or discussion section and explain why the increase in LC3 puncta in macrophages from AKI mice was less than in NC mice.

h. In Figure 2c, the TNF α western blot bands appear unclear and connected, which compromises the accuracy of quantification. Please provide a clearer representative blot and reevaluate the quantification for accuracy.

i. It is crucial that the authors improve the organization of the figures. For example, in Figure 2, the M1/M2 ratio bar graph should be placed next to Figure 2b for better clarity, rather than next to Figure 2c. Also, in figure 3, the quantification of KIM1 area should be placed adjacent to representative images of immunohistochemistry staining of KIM1, not next to H&E staining. Similarly, Tunnel quantification should be placed adjacent to representative micrographs showing Tunnel staining. I encourage the authors to thoroughly review the arrangement of all figures to enhance the overall clarity and presentation of the data.

j. The authors are recommended to quantify mitochondrial ROS in Figure 5e rather than only presenting representative images.

k. In Figure 6i, the actin western blot bands appear overexposed and connected. Please provide a clearer representative blot and reevaluate the quantification for accuracy.

l. It is crucial that the authors display all statistical significances between the groups, particularly in Figure 6e and 6f. This is important for the credibility and robustness of the study's findings.

m. In Figure 7d, the labeling of the bar graph might be incorrect. Could the authors please verify if the dark purple bar corresponds to EXO2+sisirt3+Anti-miR-195a-5p.

n. In Figure 8f, it is recommended that the authors quantify the TUNEL staining instead of only showing representative images.

o. In the figure legend of Figure S3, GM130 is described as an exosomal marker. Please make sure to correct that to a non-exosomal marker.

p. In Figure S5, the western blot bands of NDUF54 and Actin are overexposed and connected. Please provide a clearer representative blot and reevaluate the quantification.

q. Please provide quantification of KIM1 immunostaining in Figure S8 a.

r. It is essential for the authors to provide uncropped western blot membranes and thoroughly review all western blots, ensuring that bands are neither overexposed nor connected to maintain the accuracy of quantification.

Minor Comments

- a. It would enhance the overall flow and readability of the title if the authors consider modifying it to “Autophagy Deficiency in Macrophages Exacerbates Cisplatin-Induced Mitochondrial Dysfunction and Kidney Injury via miR-195a-5p-SIRT3 axis”.
- b. In line 175, the authors refer to figures 6e and g when discussing the protein levels of ETC complexes. Please note that only figure 6e should be referenced. Figure 6g pertains to representative images of H&E, PAS staining, and immunohistochemistry staining ATP5b and KIM1.
- c. In the figure legend of Figure 5, line 803, the abbreviation for tubular epithelial cells should be TECs and not TCEs.

Reviewer #2 (Remarks to the Author):

Yuan et al studied the role of macrophage autophagy in acute kidney injury (AKI) and related mechanisms. They found that macrophage autophagy was activated and then inhibited in cisplatin-induced AKI mice. Macrophage-specific depletion of ATG7 (Atg7 Δ mye) aggravated kidney injury in AKI mice, which might be mediated via macrophage-derived exosomes, and a miR-195a-5p-SIRT3 axis involved. Overall, this is a good study to explore the role of macrophages and mechanisms in AKI and opens potential new targets for AKI treatments. However, some major issues need to be fixed.

1. Multiple Atgs often work as a complex in mitophagy, such as Atg16-Atg-12-Atg5 or Atg2-18. Thus, I am wondering if depletion of ATG7 only is sufficient for mitophagy deficiency. Furthermore, there were no data showed macrophage-derived exosomes carry Atg7 in their cargo, and if they are not inside exosomes, then Atg9 is the only one on membrane. These are essential for the study needed to be sorted out.
2. Some techniques need to be verified. Page15, miR-195a-5p in vivo transfection requires specific reagents. Page16 CM-Dil-labeled TEC is in red, PKH26 is also red.
3. The limitations of the study are missing.
4. Mitochondria autophagy is defined as mitophagy, which I prefer to use for this paper.

Reviewer #3 (Remarks to the Author):

In the present work, Yuan et al study the communication between macrophages and kidney tubular epithelial cells through exosomes in the context of cisplatin-induced acute kidney injury (AKI). Authors investigate the role of autophagy in macrophages in this context, which caused an upregulation of miR-195a-5p secretion in exosomes. Elevated miR-195a-5p would exacerbate acute kidney injury by the downregulation of Sirt3. As consequence, inhibition of this miRNA or overexpression of Sirt3 improved kidney function upon AKI. Finally, depletion of macrophages and reconstitution with AKI-derived macrophages in which autophagy was induced with trehalose reduced AKI-associated kidney injury.

While the results are interesting and consistent, there are still several concerns:

-A critical point for the claiming of kidney macrophages to TECs communication in vivo is to demonstrate that, indeed, kidney macrophages do what authors have shown for peritoneal macrophages. Most of the experiment employ peritoneal macrophages, and authors extrapolate their results to kidney macrophages. However, this is an assumption that need more validation, especially considering the heterogeneity of macrophages depending on their origin. How similar are peritoneal macrophages to kidney macrophages? Do kidney macrophages also upregulate miR-195a-5p when mice are treated with Cisplatin?

-Along the same line and due to macrophage heterogeneity, LysM is not equally active in all tissue macrophages (doi: 10.1007/978-1-4939-7837-3_24). The authors need to confirm LysM-Cre mediated recombination in kidney macrophages. By the way, In Figure S2 it seems there is mislabelling of the WT and floxed bands.

-In line with the previous point and despite there is a clear effect due to the lack of Atg7 in macrophages, authors never demonstrated that deletion of Atg7 is associated to impaired autophagy. They should induce autophagy in these cells and show whether classical autophagy markers are induced.

-Authors definitely use extremely high concentrations for many experiments. For instance, 50-100 ug exosomes are too much, considering that circulating exosomes are considered to be 30 ug aprox. Treating TECs cells with so many exosomes could lead to unspecific effects. I would like to see whether the effects are maintained when used exosomes from macrophages treated with anti-miR-195a inhibitor.

-Similarly, dose of mimic miR-195a-5p is too high. If effect on Sirt3 is really specific they could see downregulation of the target at much lower doses. Indeed, authors need to show the Sirt3 prediction. Other Sirt proteins are also targeted by miR-195a-5p, such as Sirt1, which is among the experimentally-validated targets for this microRNA. Could the authors check whether part of the phenotype is due to changes in other Sirt proteins?

-Authors claim that "EXO1 had minimal effects on renal function and the severity of the AKI" (page 6). What is the base for that claim? Authors did not include a non-exo treatment control for the experiment in which they inject EXO1 and EXO2. It would have been an appropriate control for that experiment.

-I am missing quantification of total exosomal numbers by Nanoparticle tracking analysis (NTA). Authors only included protein quantification and size/concentration curve. Besides, they would need to represent these size/concentration curve in the same graph, as it seems EXO2 are generally smaller.

-Apoptosis: In Fig. S3, authors plot a representative FACS gating. However, they never mentioned which condition is taken as apoptotic cells (I guess Annexin V+ PI-). Necrosis is also induced (both markers +). Besides, none of the presented individual dots in S3g match with the representative FACS image.

-It was not clear to me the dynamics of macrophage accumulation in the context of cisplatin administration. There was an induction of autophagy marker LC3 at day 2nd, but decrease at day 4th. When was the macrophage characterization (Figure 2) done: day 2 or 4? In this same figure, I would like to see side-by-side the representative FACS of both control and Atg7 knock-down macrophages?

-Reference for trehalose to induce autophagy is missing.

-Error in legend fig. 7d

Responses to the reviewer's comments

Reviewer #1 (Remarks to the Author):

The manuscript by Yuan et al. reveals that M ϕ autophagy is initially activated and then inhibited in cisplatin-induced AKI mice. The authors demonstrate that the M ϕ -specific deletion of ATG7 exacerbates kidney injury, leading to tubulointerstitial inflammation in cisplatin-induced AKI mice. Importantly, the authors illustrate the involvement of M ϕ -derived exosomes from Atg7 Δ mye mice, particularly miR-195a-5p, in impairing mitochondria in tubular epithelial cells by interacting with SIRT3. Moreover, the authors demonstrate that inhibiting miR-195a-5p and overexpressing SIRT3 enhance mitochondrial function and improve renal function in cisplatin-induced Atg7 Δ mye AKI mice. Finally, the authors show that adoptive transfer of M ϕ from AKI mice to M ϕ -depleted mice exacerbates the kidney injury response to cisplatin. However, this effect is mitigated when M ϕ autophagy is activated with trehalose.

2. Comments to the Authors

The study by Yuan et al. pertains to a crucial area of research. The authors highlight M ϕ autophagy as a crucial contributor to AKI progression and identify miR-195a-5p/SIRT3 axis as a potential therapeutic target. However, there are several major and crucial points that need to be addressed to meet the standards of Nature Communications and engage its readers.

Major Comments

1. To better understand the role of macrophages in AKI, the authors are encouraged to conduct experiments involving macrophages depletion. This approach will enable them to determine the specific contribution of macrophages, with a particular focus on autophagy, in injury during AKI.

Answer: Thanks for your valuable suggestions. According to the suggestions of the reviewer, to better understand the specific role of macrophages (M ϕ), with a particular focus on autophagy, in injury during AKI, we conducted the experiments involving M ϕ depletion in cisplatin-treated Atg7 Δ mye mice. Results showed that after clodronate liposomes (Lipo-Clod) injection, kidney-infiltrated M ϕ was significantly depleted as determined by flow cytometry (Fig. 1a). Compared to the findings in cisplatin-induced Atg7 Δ mye mice, the serum BUN and CREA levels were decreased in Lipo-Clod treated mice (Fig. 1b, c). The kidney morphologic abnormalities and tubular damage score were ameliorated after M ϕ depletion (Fig. 1d). Collectively, these results indicated that M ϕ depletion protected against cisplatin-induced kidney injury in Atg7 Δ mye mice.

Fig. 1. Macrophages (M ϕ) depletion protected against cisplatin-induced kidney injury in Atg7^{Δmye} mice. (a) Flow cytometry analysis for M ϕ depletion (n=3). (b and c) The serum levels of BUN and CREA in Atg7^{Δmye} mice (n=6). (d) Representative images of hematoxylin-eosin (HE) and periodic acid-Schiff (PAS)-stained kidney sections, and the tubular injury score in mice (n=4). Scale bars, 50 μ m. Data are means \pm SEM. **P< 0.01.

2. It is essential to measure M1 macrophage markers in addition to CD11c, including CD86, iNOS, Fas ligand, as well as the M2 macrophage markers including CD163 and Arg-1. A panel of markers is often employed to characterize their polarization states accurately.

Answer: Thanks for your suggestions, it's necessary to employ a panel of markers to accurately

characterize the polarization states of M ϕ . According to your suggestions, we measured M1 M ϕ markers (CD86, iNOS and Fas ligand (FasL)), as well as the M2 M ϕ markers (CD163 and Arg1) by RT-PCR. Consistent with the results measured by flow cytometry, phenotype of M1 M ϕ in the kidney tissues were observed a significant increase in Atg7 Δ mye mice after cisplatin injection. In addition, the expression levels of CD163 and Arg1 were no obvious change (Fig. 2).

Fig. 2. The measurement of M1 and M2 M ϕ markers. Phenotype of M ϕ in the kidney tissues was characterized by RT-PCR (M1: CD86, iNOS, FasL; M2: CD163 and Arg1) (n=3). Data are means \pm SEM. *P<0.05, **P<0.01.

3. It is also essential to assess the effect of chemokines on macrophage polarization in AKI.

Answer: Thanks for your nice comment on our article. Inflammatory chemokines are major players in the process of recruitment and activation in AKI. Distinct chemokine repertoires associate with macrophages (M ϕ) polarization¹. Therefore, it is essential to assess the effect of chemokines on M ϕ polarization in AKI. Firstly, we detected the expression of chemokines in kidney from WT and Atg7 Δ mye mice after cisplatin injection. Compared to WT mice, we observed a significant increase of the levels of CCL5, CCL8, CXCL3, and CCL2 in the kidney tissues of Atg7 Δ mye mice after cisplatin injection (Fig. 3a). To further assess the effect of chemokines on M ϕ polarization in AKI, M ϕ from WT (M ϕ ^{WT}) and Atg7 Δ mye (M ϕ ^{Atg7 Δ mye}) mice were incubated with recombinant chemokines. M1 markers (CD86 and FASL) were significantly up-regulated in M ϕ ^{WT} after the treatment of recombinant chemokines CCL2 and CCL5, and autophagy deficiency promoted a further increase of the expression of M1 markers in M ϕ ^{Atg7 Δ mye} (Fig. 3b, c). Moreover, the expression of M2 markers (Arg1 and CD163) were inhibited after the treatment of recombinant chemokine CCL8 (Fig. 3d). However, no significant effect on M ϕ polarization was observed after incubation with recombinant CCL3 (Fig. 3e).

Fig. 3. The effect of chemokines on macrophages (M ϕ) polarization in AKI. (a) The mRNA level of chemokines in the kidney tissues of WT and Atg7 Δ mye mice after cisplatin injection. (b-e) The mRNA level of M1 and M2 markers were determined after incubation with recombinant chemokines (CCL5, CCL8, CXCL3, and CCL2) in M ϕ ^{WT} and M ϕ ^{Atg7 Δ mye}. Data are means \pm SEM. *P < 0.05, **P < 0.01.

4. In the introduction, the authors referenced a study demonstrating that the loss of mitophagy proteins PINK1 and Parkin exacerbated macrophage-derived kidney fibrosis. Therefore, it is highly encouraged for the authors to assess the levels of Parkin and PINK1 proteins when evaluating mitochondrial function. This will help elucidate whether Atg7 Δ mye mice exhibit mitochondrial dysfunction as a result of impaired mitophagy.

Answer: We sincerely appreciate the valuable comments. It is necessary to elucidate whether Atg7 Δ mye mice exhibit mitochondrial dysfunction as a result of impaired mitophagy. Our previous research, as well as other investigators, demonstrated that the mitophagy process in kidney tissue was activated in AKI induced by cisplatin², contrast³ and septic⁴, further activation of mitophagy could ameliorate mitochondrial damage. Consistent with the previous researches, mitophagy was activated in the kidney of cisplatin-treatment AKI, which was confirmed by the increased protein expression levels of PINK1, Parkin and LC3 II. However, there was no significant differences of mitophagy-related proteins between WT and Atg7 Δ mye mice under AKI condition, which indicated that mitochondrial dysfunction in cisplatin-induced Atg7 Δ mye mice may independent on mitophagy. (Fig. 4).

Fig. 4. The expression of mitophagy-related protein in cisplatin-induced Atg7 Δ mye mice. The

protein level of Parkin, PINK1 and LC3 II in kidney tissue from WT and *Atg7^{Δmye}* mice after cisplatin injection. ACTIN was used as the loading control (n=3). Data are means \pm SEM. ****P**< 0.01.

5. In Figure 1c, could the authors please specify if the bar graph represents LC3-II/LC3-I or LC3-II normalized to the loading control actin.

Answer: We feel really sorry that the description is not clear in the paper. The bar graph represents LC3 II/ACTIN, and we have noted “ACTIN was used as the loading control” in the figure legend.

6. The authors mentioned a dramatic decrease in the percentage of M ϕ with LC3-II puncta on the 4th day after cisplatin treatment (Fig. 1d). Could the authors please provide quantification for LC3 staining in peritoneal M ϕ after treatment with hydroxychloroquine (HCQ, 20 μ M) following cisplatin injection for 4 days to justify their statement?

Answer: According to your suggestion. We have added the quantification for LC3 staining in peritoneal M ϕ after treatment with hydroxychloroquine (HCQ, 20 μ M) following cisplatin injection for 4 days in the revised manuscript (Fig. 5).

Fig. 5. Immunofluorescence of LC3 in peritoneal M ϕ . Representative images and quantification of LC3 staining in peritoneal M ϕ with the treatment of hydroxychloroquine (HCQ, 20 μ M) after cisplatin injection for 4 d. Scale bar, 20 μ m. Data are means \pm SEM. ****P**< 0.01.

7. In Figure 1d, the authors utilized hydroxychloroquine, an autophagy inhibitor, to assess LC3 puncta. It is crucial to discuss and comment on autophagic flux in the results or discussion section and explain why the increase in LC3 puncta in macrophages from AKI mice was less than in NC mice.

Answer: We greatly appreciate for your professional advices. In our research, the level of LC3 II in the peritoneal macrophages (M ϕ) were increased and the level of P62 was decreased on the 2th day after cisplatin injection, suggesting autophagy was activated, while the expression of LC3 II were quickly dropped to lower level than NC mice on the 4th day after cisplatin treatment (Fig. 6a). The expression of LC3 II could be influenced due to the mature of autophagosomes and the activity of autophagic flux. To further analyze the cause of decreased LC3 II expression, M ϕ were treated with lysosomal inhibitors hydroxychloroquine (HCQ), which inhibits autophagic hyperactivity by blocking autophagosome fusion and degradation. According to the guidelines for the use and interpretation of assays for monitoring autophagy (4th edition)⁵, Autophagic flux index (defined as the proportion of the levels of LC3-II or LC3 puncta in the presence of HCQ to that in the absence of HCQ) is calculated at the indicated times. In the present study, with the treatment of HCQ, the

fold change of LC3 puncta in M ϕ from AKI mice (M ϕ ^{AKI}) was less than that in NC mice, indicating the activity of autophagic flux was inhibited in M ϕ from AKI mice (Fig. 6b). In addition, the immunofluorescence of LC3, after the treatment with HCQ to inhibit the autophagic flux activity, can reflect the relative rates of autophagosome formation. Our results showed that, the MFI of LC3 in M ϕ ^{AKI} was lower comparing with M ϕ ^{NC} in HCQ group, suggesting the autophagosome formation was reduced in M ϕ ^{AKI} (Fig. 6b). Collectively, these results confirmed that M ϕ autophagy activity was decreased in the process of AKI.

Fig. 6. Autophagy activity of macrophages (M ϕ) is activated and then inhibited in cisplatin-induced AKI mice. (a) After cisplatin injection, peritoneal M ϕ were isolated, and the protein levels of autophagy genes were measured (n=3). (b) Representative images of LC3 staining in peritoneal M ϕ with the treatment of hydroxychloroquine (HCQ, 20 μ M) after cisplatin injection for 4 d. Scale bar, 20 μ m. ACTIN was used as the loading control. Data are means \pm SEM. *P<0.05, **P<0.01.

8. In Figure 2c, the TNF α western blot bands appear unclear and connected, which compromises the accuracy of quantification. Please provide a clearer representative blot and reevaluate the quantification for accuracy.

Answer: Thanks for your suggestion. To better to reflect the accuracy of quantification, we have redone the western blot experiment. Here, we have provided the representative blot (TNF- α and ACTIN) and reevaluate the quantification for accuracy in Figure 2c in the revised manuscript. The western blot bands marked in red were modified.

Fig. 7. M ϕ -specific deficient of ATG7 aggravates inflammatory response in AKI mice. The protein levels of inflammation-related genes (IL-1 β , TNF- α) were measured by western blot (n=5). ACTIN was used as the loading control. Data are means \pm SEM. **P<0.01.

9. It is crucial that the authors improve the organization of the figures. For example, in Figure 2, the M1/M2 ratio bar graph should be placed next to Figure 2b for better clarity, rather than next to Figure 2c. Also, in figure 3, the quantification of KIM1 area should be placed adjacent to representative images of immunohistochemistry staining of KIM1, not next to H&E staining. Similarly, Tunnel quantification should be placed adjacent to representative micrographs showing Tunnel staining. I encourage the authors to thoroughly review the arrangement of all figures to enhance the overall clarity and presentation of the data.

Answer: We thank the reviewer for the suggestions. To enhance the overall clarity and presentation of the data, we have reviewed and changed the arrangement of all figures in the revised manuscript.

10. The authors are recommended to quantify mitochondrial ROS in Figure 5e rather than only presenting representative images.

Answer: According to the suggestion, we have provided the quantification of mitochondrial ROS.

Fig. 8. The quantification analysis for mitochondrial ROS. Representative micrographs and the quantification analysis showing mitochondrial ROS (mtROS) in TECs. Scale bar, 10 μ m. Data are means \pm SEM. **P<0.01.

11. In Figure 6i, the actin western blot bands appear overexposed and connected. Please provide a clearer representative blot and reevaluate the quantification for accuracy.

Answer: According to the suggestion, based on the raw figures under short exposure and long exposure time, we have changed the actin band, and reevaluated the quantification (Fig. 9). The western blot bands marked in red were modified.

Fig. 9. miR-195a-5p antagomir attenuates kidney injury in cisplatin-induced *Atg7 Δ mye* mice. (a) A clearer representative blot and reevaluate the quantification of ACTIN. (n=3). (b) The raw figures of ACTIN under short exposure and long exposure time. ACTIN was used as the loading control. Data are means \pm SEM. *P<0.05, **P<0.01.

12. It is crucial that the authors display all statistical significances between the groups, particularly in Figure 6e and 6f. This is important for the credibility and robustness of the study's findings.

Answer: Thanks for your suggestion. To better reflect the credibility and robustness of our study's findings, we have displayed all statistical significances between the groups in Figure 6 in the revised manuscript.

13. In Figure 7d, the labeling of the bar graph might be incorrect. Could the authors please verify if the dark purple bar corresponds to EXO2+sisirt3+Anti-miR-195a-5p.

Answer: I apologize for the errors in the paper, the dark purple bar corresponds to EXO2+sisirt3+Anti-miR-195a-5p. And we have corrected the labeling in the Figure 7d.

14. In Figure 8f, it is recommended that the authors quantify the TUNEL staining instead of only showing representative images.

Answer: According to the suggestion. We have provided the quantification of the TUNEL staining in the revised manuscript (Fig. 10).

Fig. 10. The quantification of the TUNEL staining. Representative micrographs and the quantification analysis showing the TUNEL staining in each group (line box indicates the magnified image) (n=6). Scale bar, 50 μ m and 10 μ m. Data are means \pm SEM. *P< 0.05, **P<0. 01.

15. In the figure legend of Figure S3, GM130 is described as an exosomal marker. Please make sure to correct that to a non-exosomal marker.

Answer: We feel really sorry for that. We have rewritten the figure legend of Figure S3 as “Western blot analysis of exosomal positive markers (CD9, Alix) and negative markers (GM130)”.

16. In Figure S5, the western blot bands of NDUFS4 and Actin are overexposed and connected.

Please provide a clearer representative blot and reevaluate the quantification.

Answer: According to the suggestion, based on the raw figures under short exposure and long exposure time, we have changed the NDUFS4 and ACTIN western blot band, and reevaluated the quantification (Fig. 11). The western blot bands marked in red were modified.

Fig. 11. Overexpression of SIRT3 alleviates cisplatin-induced the decreased of OXPHOS-related proteins. (a) A clearer representative blot and reevaluate the quantification of NDUFS4 and ACTIN. (n=3). (b) The raw figures of NDUFS4 and ACTIN under short exposure and long exposure time. ACTIN was used as the loading control. Data are means \pm SEM. *P< 0.05, **P< 0.01.

17. Please provide quantification of KIM1 immunostaining in Figure S8 a.

Answer: According to your suggestion. We have provided the quantification of KIM1 immunostaining in the revised manuscript (Fig. 12).

Fig. 12. Adoptive transfer of autophagy-activated macrophages (M ϕ) reduces KIM1 expression. Immunohistochemistry staining and quantification of KIM1 expression in kidney

sections (n=4). Scale bar, 50 μ m. Data are means \pm SEM. *P< 0.05, **P< 0.01.

18. It is essential for the authors to provide uncropped western blot membranes and thoroughly review all western blots, ensuring that bands are neither overexposed nor connected to maintain the accuracy of quantification.

Answer: Thank for your suggestion. We have provided all uncropped western blot membranes in the file of source data. Moreover, we have thoroughly checked and changed the all overexposed or connected western blot bands, and reevaluated the accuracy of quantification in the revise manuscript.

Minor Comments

1. It would enhance the overall flow and readability of the title if the authors consider modifying it to “Autophagy Deficiency in Macrophages Exacerbates Cisplatin-Induced Mitochondrial Dysfunction and Kidney Injury via miR-195a-5p-SIRT3 axis”.

Answer: We sincerely appreciate the valuable suggestion. To enhance the overall flow and readability of the title, we have modified the title to “Autophagy Deficiency in Macrophages Exacerbates Cisplatin-Induced Mitochondrial Dysfunction and Kidney Injury via miR-195a-5p-SIRT3 axis”.

2. In line 175, the authors refer to figures 6e and g when discussing the protein levels of ETC complexes. Please note that only figure 6e should be referenced. Figure 6g pertains to representative images of H&E, PAS staining, and immunohistochemistry staining ATP5b and KIM1.

Answer: Thanks for your suggestion. We have changed the reference to figure 6e when discussing the protein levels of ETC complexes in the revised manuscript.

3. In the figure legend of Figure 5, line 803, the abbreviation for tubular epithelial cells should be TECs and not TCEs.

Answer: Thanks for your careful checks. We feel sorry for our carelessness. We have corrected the mistake in the revised manuscript.

Reviewer #2 (Remarks to the Author):

Yuan et al studied the role of macrophage autophagy in acute kidney injury (AKI) and related mechanisms. They found that macrophage autophagy was activated and then inhibited in cisplatin-induced AKI mice. Macrophage-specific depletion of ATG7 (Atg7 Δ mye) aggravated kidney injury in AKI mice, which might be mediated via macrophage-derived exosomes, and a miR-195a-5p-SIRT3 axis involved. Overall, this is a good study to explore the role of macrophages and mechanisms in AKI and opens potential new targets for AKI treatments. However, some major issues need to be fixed.

1. Multiple Atgs often work as a complex in mitophagy, such as Atg16-Atg-12-Atg5 or Atg2-18. Thus, I am wondering if depletion of ATG7 only is sufficient for mitophagy deficiency. Furthermore, there were no data showed macrophage-derived exosomes carry Atg7 in their cargo, and if they are

not inside exosomes, then Atg9 is the only one on membrane. These are essential for the study needed to be sorted out.

Answer: Thanks for your valuable suggestions. Autophagy is an essential cellular degradation process that delivers cytoplasmic components to lysosomes, which is associated with a series of multiple ATGs, such as ATG16-ATG-12-ATG5 or ATG2-18. During the process, autophagosome formation requires many ATGs, especially ATG5 (substrate), ATG7, ATG3, and ATG10 (enzymes) are critical for ligation with PE⁶. It is essential for the study to be sorted out if depletion of ATG7 only is sufficient for autophagy deficiency. Consistent with the reports by Guo⁷ and Choi⁸, we also observed that along with the knockout of ATG7 in *Atg7^{Δmye}* mice, the expression of LC3 II was reduced, while the expression of P62 was increased, indicating that the autophagy in macrophages (Mφ) was impaired (Fig. 13a).

Existing literature has suggested contradicting effects of autophagy genes on exosome production. Guo et al have demonstrated that deficiency of ATG5 but not ATG7 reduced the exosome released via detaches ATP6V1E1 from the V1V0-ATPase⁹. In addition, Murrow et al have revealed that ATG7 is s required for the formation of ATG3-12 complex, which could promote exosome release¹⁰. While this contradicts a previous report that ATG7 makes no difference on exosome release¹¹. Here, we found that in comparison with Mφ-derived exosomes from WT mice (Mφ^{WT}-EXO, EXO1), the levels of exosome markers (CD9 and Alix) and exosome-sized particles were not significantly changed in exosomes from equivalent amounts of Mφ from Mφ^{*Atg7^{Δmye}*} mice (Mφ^{*Atg7^{Δmye}*}-EXO, EXO2) (Fig. 13b, c), indicating that ATG7 has no impact on exosome release in Mφ. In addition, Guo et al also found that exosomes-derived from mouse embryonic fibroblasts (MEFs) could not detect the expression of ATG7⁹. Similar with the observation, our results showed that Mφ-derived exosomes did not carry ATG7. ATG9 is the only transmembrane protein known to be involved in autophagy, and is required to form intraluminal vesicles¹². Though, ATG9 can be detected on small 30–60 nm vesicles at the pre-autophagosome structure (PAS)¹³, our study found that ATG9 is not inside exosomes (Fig. 13d).

Fig. 13. Myeloid ATG7 was efficiently deleted in *Atg7^{Δmye}* mice and the characterization of exosomes. (a) The protein level of ATG7, P62, and LC3 II in Mφ from WT and *Atg7^{Δmye}* mice (n=3). **(b)** Immunoblot analysis of exosomal positive markers (CD9, Alix) and negative markers (GM130). **(c)** The exosomal size and particles of EXO1 and EXO2 were measured by NTA (n=3). **(d)**

Representative images of western blot of ATG7, ATG9, CD9 and GM130. Data are means \pm SEM. ** $P < 0.01$. EXO1, M ϕ ^{WT}-EXO; EXO2, M ϕ ^{Atg7 Δ mye}-EXO

2. Some techniques need to be verified. Page15, miR-195a-5p *in vivo* transfection requires specific reagents. Page16 CM-Dil-labeled TEC is in red, PKH26 is also red.

Answer: Thanks for your suggestions. According to the manufacturer's instructions of GenePharma (Shanghai, China), compared to common miRNA inhibitor, antagomir has a higher affinity with cell membrane, transfection reagent is not necessary to be used in *in vivo* animal experiment. One research showed that antagomiR-145-5p and antagomir-NC were directly injected into the buccal and palatal sides of the maxillary without the coordination of transfection reagent¹⁴. In the researches of Xu and Zuo, miR-320e antagomir and miR-7 antagomir were administrated to mice through tail vein, which was not involved transfection reagent^{15 16}. In the present study, miR-195a-5p antagomir was directly administered to Atg7 Δ mye mice through tail vein 24 h before cisplatin injection without specific transfection reagents. In addition, I apologize for the errors of wrong spelling about PKH26 in the manuscript. CM-Dil-labeled TEC is in red and PKH67-labeled exosomes is in green in our experiment. And we have corrected this mistake in the revised manuscript.

3. The limitations of the study are missing.

Answer: Our study highlights M ϕ autophagy as a crucial contributor to AKI progression and identify miR-195a-5p/SIRT3 axis as a potential therapeutic target. However, there stills several limitations. Firstly, ATG7 could exert autophagy-independent effects. Yao et al demonstrated that ablation of endothelial ATG7 could inhibit ischemia-induced angiogenesis by upregulating Stat1 to suppress Hif1 α expression¹⁷. Moreover, the absence of ATG7 resulted in augmented DNA damage with increased p53-dependent apoptosis during metabolic stress¹⁸. Though our study has demonstrated that the impaired M ϕ autophagy is crucial to the aggravated kidney injury in Atg7 Δ mye mice, ATG7-mediated autophagy-independent mechanism may contribute to cisplatin-induced kidney injury in AKI mice¹⁹. Secondly, Lyz2 promoter is known to confer the efficient activity not only in monocytes/ M ϕ but also in other cells of myeloid origin under steady-state conditions, such as neutrophils, and eosinophils⁸. Although M ϕ depletion significantly protected against cisplatin-induced kidney injury in Atg7 Δ mye mice, the further investigation whether ATG7 of the cell subset also contributes to the kidney injury in Atg7 Δ mye mice is necessary. Lastly, it would be more convincing if our experiments were verified on different AKI model such as ischemia-reperfusion model.

4. Mitochondria autophagy is defined as mitophagy, which I prefer to use for this paper.

Answer: Thanks for your suggestion. Mitochondria autophagy is defined as mitophagy. To elucidate whether Atg7 Δ mye mice exhibit mitochondrial dysfunction as a result of impaired mitochondria autophagy, we have detected mitochondria autophagy process in kidney tissue, it is more appropriate to change mitochondria autophagy to mitophagy in this section. In addition, we generated the myeloid cell-specific Atg7-deficient (Atg7 Δ mye) mice. Based on the regulatory role of ATG7 in canonical macroautophagy, M ϕ autophagy may be more proper than mitophagy in this section of this manuscript.

Reviewer #3 (Remarks to the Author):

In the present work, Yuan et al study the communication between macrophages and kidney tubular epithelial cells through exosomes in the context of cisplatin-induced acute kidney injury (AKI). Authors investigate the role of autophagy in macrophages in this context, which caused an upregulation of miR-195a-5p secretion in exosomes. Elevated miR-195a-5p would exacerbate acute kidney injury by the downregulation of Sirt3. As consequence, inhibition of this miRNA or overexpression of Sirt3 improved kidney function upon AKI. Finally, depletion of macrophages and reconstitution with AKI-derived macrophages in which autophagy was induced with trehalose reduced AKI-associated kidney injury.

While the results are interesting and consistent, there are still several concerns:

1. A critical point for the claiming of kidney macrophages to TECs communication in vivo is to demonstrate that, indeed, kidney macrophages do what authors have shown for peritoneal macrophages. Most of the experiment employ peritoneal macrophages, and authors extrapolate their results to kidney macrophages. However, this is an assumption that need more validation, especially considering the heterogeneity of macrophages depending on their origin. How similar are peritoneal macrophages to kidney macrophages? Do kidney macrophages also upregulate miR-195a-5p when mice are treated with Cisplatin?

Answer: We gratefully thanks for your professional review work. The origin of macrophages (M ϕ) in the kidney and peritoneal M ϕ is different. The heterogeneity and inherently plastic of M ϕ depend their origin and may adopt different phenotypes in response to environmental cues²⁰. We isolated and kidney M ϕ (KM) and peritoneal M ϕ (PM) from normal and AKI mice, and determined the mRNA expression of inflammation-related factors and chemokines in PM and KM. The results showed that, compared to NC group, PM and KM derived from AKI mice exhibited similar phenotypes, characterized by the increased expression of M1 markers and chemokines. However, there was no significant differences on M2 markers (Fig. 14a). Moreover, the levels of pro-inflammatory related factors in KM were higher than that in PM under AKI condition, suggesting that KM may have stronger immune response capacity (Fig. 14a). An accumulation of evidence has revealed the intrinsic interaction between phagocytosis and M ϕ -mediated inflammation²¹. Thus, we evaluate the phagocytosis function in PM and KM. Briefly, PM and KM were incubated with pHrodo Green E. coli BioParticles (Thermo, P35366) for 90 min at 37 °C. Immunofluorescence results showed that the phagocytic ability of KM was significantly higher than the PM in the NC mice (Fig. 14b). After cisplatin injection, the phagocytic ability of KM and PM were both decreased. More importantly, the miR-195a-5p level in KM and PM from AKI mice were enriched than that in NC mice (Fig. 14c). Above all, although the origin of KM and PM is different, they share some similarity in AKI mice.

Fig. 14. The characteristic evaluation of peritoneal macrophages (PM) and kidney macrophages (KM). (a) The mRNA levels of M1, M2 markers and chemokines were measured. (b) Fluorescence images of PM and KM incubated with pHrodo Green *E. coli* BioParticles (green). Scale bars, 20 μ m. (c) The level of miR-195a-5p in PM and KM from NC and AKI mice (n=3). Data are means \pm SEM. *P< 0.05, **P< 0.01.

2. Along the same line and due to macrophage heterogeneity, LysM is not equally active in all tissue macrophages (doi: 10.1007/978-1-4939-7837-3_24). The authors need to confirm LysM-Cre mediated recombination in kidney macrophages. By the way, In Figure S2 it seems there is misslabelling of the WT and floxed bands.

Answer: Thanks for your suggestion. Actually, LysM is not equally active in all tissue macrophages (M ϕ)²². In our study, Atg7^{fl/fl} mice with floxed alleles for the autophagy gene Atg7 were crossed with Lyz2-Cre mice with the mouse lysozyme M promoter-driven Cre recombinase to generate myeloid cell specific deletion of Atg7 mice (Atg7 ^{Δ mye}) (Fig. 15a, b). To confirm LysM-Cre mediated recombination in our research, we measured the expression of autophagy-related genes in KM and PM respectively. The results showed that the high efficiency of LysM-Cre mediated recombination was measured in KM and PM in Atg7 ^{Δ mye} mice (Fig. 15c-e). In order to describe the results of genotyping more accurately and clearly, we have modified Fig.S2b as following (Fig. 15b). In addition, the WT and floxed bands are not mislabeled, we attached a genotype PCR results provided by RIKEN BRC in Fig. 15f.

Fig. 15. Macrophages (M ϕ)-specific depletion of ATG7 in $Atg7^{\Delta mye}$ mice. (a) Experimental scheme for generating the $Atg7^{\Delta mye}$ mice. (b) Phenotype identification of $Atg7^{\Delta mye}$ mice by PCR. (c) The $Atg7$ mRNA level in kidney M ϕ (KM). (d and e) Representative images of western blot and quantitative analyses of autophagy-related genes (ATG7, P62, and LC3II) in KM and peritoneal M ϕ (PM) from WT and $Atg7^{\Delta mye}$ mice. (f) A genotype PCR results about $Atg7^{\Delta mye}$ mice provided by RIKEN BRC. ACTIN was used as the loading control (n=3). Data are means \pm SEM. **P< 0.01.

3. In line with the previous point and despite there is a clear effect due to the lack of $Atg7$ in macrophages, authors never demonstrated that deletion of $Atg7$ is associated to impaired autophagy. They should induce autophagy in these cells and show whether classical autophagy markers are induced.

Answer: According to your suggestion. To demonstrate the deletion of $Atg7$ is associated to impaired autophagy, M ϕ from WT (M ϕ^{WT}) and $Atg7^{\Delta mye}$ (M $\phi^{Atg7\Delta mye}$) mice were treated with trehalose (a natural nonreducing disaccharide), a novel autophagy inducer. The results showed that, compared to M ϕ^{WT} , autophagy was impaired in M $\phi^{Atg7\Delta mye}$, which was presented by the reduced expression of LC3 II and the increased expression of P62. Moreover, after incubation with trehalose, the expression of LC3 II, $Atg7$ and P62 was increased in M ϕ^{WT} , suggesting autophagy activation. However, the depletion of $Atg7$ blocked the effects of trehalose (Fig. 16).

Fig. 16. Autophagy was impaired in *Atg7^{Δmye}* mice. (a) Peritoneal Mφ isolated from NC and *Atg7^{Δmye}* mice were treated with trehalose, then the protein levels of LC3 II and P62 were detected (n=3). ACTIN was used as the loading control (n=3). Data are means ± SEM. **P< 0.01.

4. Authors definitely use extremely high concentrations for many experiments. For instance, 50-100 μg exosomes are too much, considering that circulating exosomes are considered to be 30 μg approx. Treating TECs cells with so many exosomes could lead to unspecific effects. I would like to see whether the effects are maintained when used exosomes from macrophages treated with anti-miR-195a inhibitor.

Answer: We agree with your suggestion. In our previous study, the cells were seeded into six wells, and treated with 50 μg exosomes (EXO), which the final concentration of EXO is 25 μg/ml. As the reviewer suggestion, the EXO used in our study are too much, in order to exclude the unspecific effects of EXO, the cells were treated with EXO in different concentrations for 48h. Here, we found that, compared with EXO1 (Mφ^{WT}-EXO), TECs exhibited decreased cell viability after treatment with EXO2 (Mφ^{Atg7Δmye}-EXO) in a concentration-dependent manner. And at a dose of 10 μg/ml EXO2, the cell viability was decreased to 0.69 ± 0.01 (Fig. 17a). Thus, we used the concentration of 10 μg/ml EXO2 for subsequent experiments. In addition, cell apoptosis was increased after EXO2 treatment (Fig. 17b).

Similar with effects of 25 μg/ml EXO2, which was used in previously, the mitochondria in TECs was impaired after the treatment with 10 μg/ml EXO2. Due to the miRNAs transfer by exosomes, a higher abundance of miR-195a-5p level in recipient TECs was also observed after the EXO2 incubation *in vitro* (Fig. 17c). Then, we further explored the effects of anti-miR-195a inhibitor under this stimuli. Here, the mitochondrial depolarization and excessive generation of mtROS induced by EXO2 at a dose of 10 μg/ml were alleviated by miR-195a-5p inhibitor (Fig. 17d, e). As we all known, mitochondrial fragmentation is associated with increased mtROS. Comparing with elongated networks of mitochondria in control TECs, mitochondria in EXO2-induced cells appeared smaller and punctate pattern, whereas the mitochondrial fragmentation caused by EXO2 was improved by miR-195a-5p inhibitor treatment (Fig. 17f). Consequently, the reduced ATP generation and suppressed OCR induced by EXO2 were partially reversed by miR-195a-5p inhibitor (Fig. 17g, h). These data suggested that miR-195a-5p inhibitor not only restored the morphology of mitochondria but also effectively improved mitochondrial bioenergetics in TECs.

Fig. 17. Macrophages (M ϕ) derived miR-195a-5p impairs mitochondria in TECs. (a) The CCK8 assay examined the cell viability of TECs after incubating with exosomes in a concentration-dependent manner. (b) Apoptosis was determined by flow cytometry. (c) The level of miR-195a-5p in TECs after the incubation with exosomes. (d) Flow cytometer analysis of the mitochondrial membrane potential ($\Delta\psi_m$) in TECs. (e) Representative images and quantification of mitochondrial ROS (mROS) in TECs. Scale bars, 10 μ m (f) Representative immunofluorescence and quantification of mitochondrial morphology in TECs, loaded with Mito-tracker-green. Scale bars, 10 μ m and 2 μ m. (g) The ATP content of TECs measured by an ATP Assay Kit, and the ATP concentration was calculated in nmol/mg protein. (h) Measurement of the mitochondrial OCR in TECs using a Mito Stress kit, and the quantification of basal respiration, ATP production, maximal respiration, and spare respiration capacity. Data show a representative of at least three independent experiments. Data are means \pm SEM. * $P < 0.05$, ** $P < 0.01$. EXO1, M ϕ ^{WT}-EXO; EXO2, M ϕ ^{Atg7 Δ Mye}-EXO.

Base on the results that miR-195a-5p specifically targets SIRT3 (Fig. 18a-c). Here, we also evaluated the role of miR-195a-5p/SIRT3 axis in TECs treated with EXO2 at a dose of 10 μ g/ml, and found that the protective role of miR-195a-5p inhibitor on mitochondria dysfunction was also

effectively blunted when the SIRT3 in TECs was knockdown under this stimuli (Fig. 18d, e).

Fig. 18. Overexpression of SIRT3 alleviates EXO2-induced mitochondrial damage. (a) Prediction of SIRT3 as a target of miR-195a-5p. (b) Dual luciferase assays of the TECs co-transfected with SIRT3 luciferase reporter (pmirGLO-SIRT3-WT, SIRT3-WT) and miR-195a-5p mimic or mutant SIRT3 luciferase reporter (pmirGLO-SIRT3-MUT, SIRT3-MUT), and the luciferase activity of cells was detected using a Dual Luciferase Assay kit. (c) Western blot and quantitative analyses of SIRT3 in TECs with the treatment of miR-195a-5p mimic. (d) Representative images and quantification of mitochondrial ROS (mtROS) and mitochondrial morphology in TECs. Scale bars, 20 μm , 10 μm and 2 μm . (e) Measurement of the mitochondrial OCR in TECs using a Mito Stress kit, and the quantification of basal respiration, ATP production, maximal respiration, and spare respiration capacity. Data are means \pm SEM. ** $P < 0.01$. EXO2, $M\phi^{\text{Atg7}\Delta\text{mye}}$ -EXO.

5. Similarly, dose of mimic miR-195a-5p is too high. If effect on Sirt3 is really specific they could see downregulation of the target at much lower doses. Indeed, authors need to show the Sirt3 prediction. Other Sirt proteins are also targeted by miR-195a-5p, such as Sirt1, which is among the experimentally-validated targets for this microRNA. Could the authors check whether part of the phenotype is due to changes in other Sirt proteins?

Answer: Thanks for your suggestion. In LPS-induced sepsis mice model, miR-195 promoted intestinal epithelial cell apoptosis via targeting SIRT1/eIF2a²³. In addition, Liu et al found that Bcl-2 is the direct target of miR-195, and inhibition of miR-195 could inhibit high glucose-induced apoptosis²⁴. To determine the specific targets of miR-195a-5p in TECs, the cells were treated with

miR-195a-5p mimic in different concentrations for 48h, and the expression of SIRT1, SIRT3 and Bcl-2 were measured. Here, miR-195a-5p mimic has no significant effect on Bcl-2 whatever lower mimic concentration (10 nm) or higher concentration (50nM). Moreover, the expression of SIRT1 was slightly reduced until at a dose of 50 nM. While, the protein level of SIRT3 was significantly decreased in a concentration-dependent manner. (Fig. 19), suggesting that miR-195a-5p specifically targeted SIRT3 in TECs.

Fig. 19. The identification of miR-195a-5p target proteins. Representative images of western blot and quantitative analyses of miR-195a-5p target proteins (Bcl-2, SIRT1, and SIRT3). ACTIN was used as the loading control (n=3). Data are means \pm SEM. *P < 0.05, **P < 0.01.

6. Authors claim that “EXO1 had minimal effects on renal function and the severity of the AKI” (page 6). What is the base for that claim? Authors did not include a non-exo treatment control for the experiment in which they inject EXO1 and EXO2. It would have been an appropriate control for that experiment.

Answer: According to your suggestion. To evaluate the effects of EXO1 on renal function and the severity of the AKI, it is necessary to add non-exo treatment control group. In our experiment, we also used NC group and cisplatin-induced AKI model as control (Fig. 20). According to the reviewer's request, we have added the data in the revised manuscript.

Fig. 20. Exosomes derived from ATG7 deficient-macrophages (M ϕ) exacerbated the kidney injury *in vivo*. Mice were injected with M ϕ -derived exosomes from WT (M ϕ ^{WT}-EXO, EXO1) or Atg7^{Ameye} mice (M ϕ ^{Atg7^{Ameye}}-EXO, EXO2) (about 100 μ g (at protein level) in 100 μ l) 24 h before cisplatin injection (16 mg/kg). (a and b) The serum levels of BUN and CREA in mice (n=6). (c and d) Representative images of hematoxylin-eosin (HE) and periodic acid-Schiff (PAS)-stained kidney

sections, and the tubular injury score in mice (n=6). Scale bars, 50 μ m. Data are means \pm SEM. *P< 0.05, **P< 0.01. NC, normal control; Cisp, cisplatin.

7. I am missing quantification of total exosomal numbers by Nanoparticle tracking analysis (NTA). Authors only included protein quantification and size/concentration curve. Besides, they would need to represent these size/concentration curve in the same graph, as it seems EXO2 are generally smaller.

Answer: Thanks for your suggestion. To evaluate whether the exosomes-mediated crosstalk between M ϕ and TECs affects kidney injury, M ϕ -derived exosomes from mice (M ϕ ^{WT}-EXO, EXO1; M ϕ ^{Atg7 Δ mye}-EXO, EXO2) were isolated and characterized by TEM, NTA, and western blot (Fig. 21a-d). The NTA analysis showed that the size of EXO2 (107.1 \pm 42.6 nm) are slightly smaller than EXO1 (120.4 \pm 58.1 nm) (Fig. 21c). And we have represented EXO1 and EXO2 size/concentration curve in the same graph. In addition, the exosomal protein and particles per million cells were calculated in EXO1 and EXO2. The results showed that exosome-sized particles were not significantly changed in EXO1 and EXO2 from equivalent amounts of M ϕ (Fig. 21e). Of note, the total exosomal protein contents per million cells was slightly increased in EXO2 compared to the EXO1 (Fig. 21f).

Fig. 21. Characterization of exosomes. The exosomes isolated from M ϕ in WT mice (M ϕ ^{WT}-EXO, EXO1) and Atg7 Δ mye mice (M ϕ ^{Atg7 Δ mye}-EXO, EXO2). (a) Schematic diagram of the methods of exosomes isolation. (b) Representative TEM micrograph of EXO1 and EXO2 (scale bar = 200 nm). (c) NTA of EXO1 and EXO2. (d) Western blot analysis of exosomal positive markers (CD9, Alix) and negative markers (GM130). (e) Exosome-sized particles and (f) total exosomal protein per million cells (n=3). Data are means \pm SEM. *P< 0.05. EXO1, M ϕ ^{WT}-EXO; EXO2, M ϕ ^{Atg7 Δ mye}-EXO.

7. Apoptosis: In Fig. S3, authors plot a representative FACS gating. However, they never mentioned which condition is taken as apoptotic cells (I guess Annexin V+ PI-). Necrosis is also induced (both markers +). Besides, none of the presented individual dots in S3g match with the representative FACS image.

Answer: Thanks for your suggestion. Previously, we take the Annexin V+ PI- and Annexin V+ PI+ cells as apoptotic cells. According to the reviewer's opinion and Nature Protocols²⁵, living cells are Annexin V- PI-, Annexin V+ PI- cells are taken as apoptotic cells, and Annexin V+ PI+ cells are

taken as necrosis. We have changed statistical graphs following the new FACS gating method and described in the methods of the revised manuscript.

8. It was not clear to me the dynamics of macrophage accumulation in the context of cisplatin administration.

Answer: According to your suggestion. We determined the kidney-infiltrated macrophages (M ϕ) respectively on the 2nd, 3rd, 4th day after cisplatin injection. The numbers of M ϕ in kidney were increased in a time-dependent manner (Fig. 22).

Fig. 22. The dynamics of macrophages (M ϕ) accumulation in AKI mice. Representative images and quantification of F4/80 in kidney sections for M ϕ detection (n=4). Data are means \pm SEM. *P<0.05, **P<0.01. NC, normal control; Cisp, cisplatin.

9. There was an induction of autophagy marker LC3 at day 2nd, but decrease at day 4th. When was the macrophage characterization (Figure 2) done: day 2 or 4? In this same figure, I would like to see side-by-side the representative FACS of both control and Atg7 knock-down macrophages?

Answer: The M ϕ characterization was determined on the 4th day after cisplatin injection. Additionally, according to the reviewer's suggestion, we have exhibited side by side the representative FASCs of both control and Atg7 knock-down M ϕ in the revised manuscript.

Fig. 23. The macrophages (M ϕ) characterization on the 4 th day in cisplatin induced AKI mice. Phenotype of M ϕ in the kidney tissues was characterized by flow cytometry (M1: CD11c+CD206-; M2: CD206+CD11c-) (n=3). Data are means \pm SEM. *P< 0.05, **P<0.01.

10. Reference for trehalose to induce autophagy is missing.

Answer: We feel sorry for our carelessness. In our revised manuscript, the reference for trehalose has been added to proper location. Thank you for your reminding.

11. Error in legend fig. 7d

Answer: We sincerely thank the reviewer for careful reading. We have corrected the mislabeled legend “EXO2+sicon+Anti-miR-195a-5p” to “EXO2+sisirt3+Anti-miR-195a-5p” in the dark purple bar of the Fig. 7d.

References

1. A. Mantovani, A. Sica, S. Sozzani, P. Allavena, A. Vecchi, M. Locati, The chemokine system in diverse forms of macrophage activation and polarization. *Trends Immunol.* **25**, 677-686 (2004).
2. L. Zhu, Y. Yuan, L. Yuan, L. Li, F. Liu, J. Liu, Y. Chen, Y. Lu, J. Cheng, Activation of TFEB-mediated autophagy by trehalose attenuates mitochondrial dysfunction in cisplatin-induced acute kidney injury. *Theranostics* **10**, 5829-5844 (2020).
3. Q. Lin, S. Li, N. Jiang, X. Shao, M. Zhang, H. Jin, Z. Zhang, J. Shen, Y. Zhou, W. Zhou *et al.*, PINK1-parkin pathway of mitophagy protects against contrast-induced acute kidney injury via decreasing mitochondrial ROS and NLRP3 inflammasome activation. *Redox Biol* **26**, 101254 (2019).
4. Y. Wang, J. Zhu, Z. Liu, S. Shu, Y. Fu, Y. Liu, J. Cai, C. Tang, Y. Liu, X. Yin *et al.*, The PINK1/PARK2/optineurin pathway of mitophagy is activated for protection in septic acute kidney injury. *Redox Biol* **38**, 101767 (2021).
5. D. J. Klionsky, A. K. Abdel-Aziz, S. Abdelfatah, M. Abdellatif, A. Abdoli, S. Abel, H. Abeliovich, M. H. Abildgaard, Y. P. Abudu, A. Acevedo-Arozena *et al.*, Guidelines for the use and interpretation of assays for monitoring autophagy (4th edition)(1). *Autophagy* **17**, 1-382 (2021).
6. H. Yamamoto, S. Zhang, N. Mizushima, Autophagy genes in biology and disease. *Nat. Rev. Genet.* **24**, 382-400 (2023).
7. L. Guo, J. Zhao, Y. Qu, R. Yin, Q. Gao, S. Ding, Y. Zhang, J. Wei, G. Xu, microRNA-20a Inhibits Autophagic Process by Targeting ATG7 and ATG16L1 and Favors Mycobacterial Survival in Macrophage Cells. *Front Cell Infect Microbiol* **6**, 134 (2016).
8. G. E. Choi, S. Y. Yoon, J. Y. Kim, D. Y. Kang, Y. J. Jang, H. S. Kim, Autophagy deficiency in myeloid cells exacerbates eosinophilic inflammation in chronic rhinosinusitis. *J. Allergy Clin. Immunol.* **141**, 938-950 e912 (2018).
9. H. Guo, M. Chitiprolu, L. Roncevic, C. Javalet, F. J. Hemming, M. T. Trung, L. Meng, E. Latreille, C. Tanese de Souza, D. McCulloch *et al.*, Atg5 Disassociates the V(1)V(0)-ATPase to Promote Exosome Production and Tumor Metastasis Independent of Canonical Macroautophagy. *Dev. Cell* **43**, 716-730 e717 (2017).
10. L. Murrow, R. Malhotra, J. Debnath, ATG12-ATG3 interacts with Alix to promote basal autophagic flux and late endosome function. *Nat. Cell Biol.* **17**, 300-310 (2015).

11. R. Sahu, S. Kaushik, C. C. Clement, E. S. Cannizzo, B. Scharf, A. Follenzi, I. Potolicchio, E. Nieves, A. M. Cuervo, L. Santambrogio, Microautophagy of cytosolic proteins by late endosomes. *Dev. Cell* **20**, 131-139 (2011).
12. C. A. Bader, T. Shandala, Y. S. Ng, I. R. Johnson, D. A. Brooks, Atg9 is required for intraluminal vesicles in amphisomes and autolysosomes. *Biol Open* **4**, 1345-1355 (2015).
13. H. Yamamoto, S. Kakuta, T. M. Watanabe, A. Kitamura, T. Sekito, C. Kondo-Kakuta, R. Ichikawa, M. Kinjo, Y. Ohsumi, Atg9 vesicles are an important membrane source during early steps of autophagosome formation. *J. Cell Biol.* **198**, 219-233 (2012).
14. X. Shen, W. Zhu, P. Zhang, Y. Fu, J. Cheng, L. Liu, R. Xu, H. Jiang, Macrophage miR-149-5p induction is a key driver and therapeutic target for BRONJ. *JCI Insight* **7**, (2022).
15. R. Zuo, L. F. Ye, Y. Huang, Z. Q. Song, L. Wang, H. Zhi, M. Y. Zhang, J. Y. Li, L. Zhu, W. J. Xiao *et al.*, Hepatic small extracellular vesicles promote microvascular endothelial hyperpermeability during NAFLD via novel-miRNA-7. *J Nanobiotechnology* **19**, 396 (2021).
16. C. Xu, Z. Zhang, N. Liu, L. Li, H. Zhong, R. Wang, Q. Shi, Z. Zhang, L. Wei, B. Hu *et al.*, Small extracellular vesicle-mediated miR-320e transmission promotes osteogenesis in OPLL by targeting TAK1. *Nat. Commun.* **13**, 2467 (2022).
17. H. Yao, J. Li, Z. Liu, C. Ouyang, Y. Qiu, X. Zheng, J. Mu, Z. Xie, Ablation of endothelial Atg7 inhibits ischemia-induced angiogenesis by upregulating Stat1 that suppresses Hif1a expression. *Autophagy* **19**, 1491-1511 (2023).
18. I. H. Lee, Y. Kawai, M. M. Fergusson, Rovira, II, A. J. Bishop, N. Motoyama, L. Cao, T. Finkel, Atg7 modulates p53 activity to regulate cell cycle and survival during metabolic stress. *Science* **336**, 225-228 (2012).
19. J. J. Collier, F. Suomi, M. Olahova, T. G. McWilliams, R. W. Taylor, Emerging roles of ATG7 in human health and disease. *EMBO Mol. Med.* **13**, e14824 (2021).
20. B. R. Conway, E. D. O'Sullivan, C. Cairns, J. O'Sullivan, D. J. Simpson, A. Salzano, K. Connor, P. Ding, D. Humphries, K. Stewart *et al.*, Kidney Single-Cell Atlas Reveals Myeloid Heterogeneity in Progression and Regression of Kidney Disease. *J. Am. Soc. Nephrol.* **31**, 2833-2854 (2020).
21. I. Kourtzelis, G. Hajishengallis, T. Chavakis, Phagocytosis of Apoptotic Cells in Resolution of Inflammation. *Front Immunol* **11**, 553 (2020).
22. J. Shi, L. Hua, D. Harmer, P. Li, G. Ren, Cre Driver Mice Targeting Macrophages. *Methods Mol. Biol.* **1784**, 263-275 (2018).
23. T. Yuan, L. Zhang, S. Yao, S. Y. Deng, J. Q. Liu, miR-195 promotes LPS-mediated intestinal epithelial cell apoptosis via targeting SIRT1/eIF2a. *Int. J. Mol. Med.* **45**, 510-518 (2020).
24. P. Liu, Q. H. Peng, P. Tong, W. J. Li, Astragalus polysaccharides suppresses high glucose-induced metabolic memory in retinal pigment epithelial cells through inhibiting mitochondrial dysfunction-induced apoptosis by regulating miR-195. *Mol. Med.* **25**, 21 (2019).
25. H. van Genderen, H. Kenis, P. Lux, L. Ungeth, C. Maassen, N. Deckers, J. Narula, L. Hofstra, C. Reutelingsperger, In vitro measurement of cell death with the annexin A5 affinity assay. *Nat. Protoc.* **1**, 363-367 (2006).

REVIEWERS' COMMENTS

Reviewer #1 (Remarks to the Author):

The authors have satisfactorily addressed all my initial comments and I do not have any further critiques to address.

Reviewer #2 (Remarks to the Author):

The revised manuscript addressed some of my concerns, added limitations, the overall quality is improved. However, my comment 1 was not fully answered. If macrophage-derived exosomes did not carry ATG7/ATG9 in WT mice, then the deficiency of ATG7 in exosomes from Atg7 depleted mice is not because of Atg7 depletion. Thus, Fig 13d should include results from WT EXO, and these should be discussed in the revision.

Reviewer #3 (Remarks to the Author):

I appreciate the efforts made by the authors to follow up all reviewer's suggestions.

I only have a minor comment: I would suggest to include exosome treatment concentration in the legends of Fig. 4 and Supp. Fig. 4i), in the same way they did for Fig. 5c.

Response to the reviewers' comments

Reviewer #1 (Remarks to the Author):

Comment 1: The authors have satisfactorily addressed all my initial comments and I do not have any further critiques to address.

Answer: We are grateful to reviewer for the valuable suggestions that have helped us improve our manuscript.

Reviewer #2 (Remarks to the Author):

Comment 1: The revised manuscript addressed some of my concerns, added limitations, the overall quality is improved. However, my comment 1 was not fully answered. If macrophage-derived exosomes did not carry ATG7/ATG9 in WT mice, then the deficiency of ATG7 in exosomes from Atg7 depleted mice is not because of Atg7 depletion. Thus, Fig 13d should include results from WT EXO, and these should be discussed in the revision.

Answer: We thank the reviewer for his/her time and constructive comments. Indeed, to investigate whether macrophages-derived exosomes carry ATG7/9 in their cargo, it would be necessary to detect the expression of ATG7/9 in exosomes from M ϕ ^{WT} and M ϕ ^{Atg7 Δ mye} (EXO1 and EXO2). Guo et al found that exosomes-derived from mouse embryonic fibroblasts (MEFs) could not detect the expression of ATG7 by Western blotting¹. However, several researches have observed that exosomes or extracellular vesicles (EVs) from myeloid-derived suppressor cells, breast cancer cells, and MEFs carry ATG7, validating by mass spectrometry²⁻⁴. Thus, we guessed that the inconsistency may be due to the low expression of ATG7 in exosomes and the differences in the sensitivity of the detection methods. ATG9 is the only transmembrane protein known to be involved in autophagy, and is required to form intraluminal vesicles, which can be detected on small 30–60 nm vesicles at the pre-autophagosome structure (PAS)⁵. In consideration of the low expression of ATG7/9 in exosomes, the total proteins used for Western blotting was up to 60 ug from 20 ug (first-revision). Comparing with the results in first-revision, both ATG7 and ATG9 inside EXO1. Moreover, due to the Atg7 depletion, the expression of ATG7 in EXO2 was significantly reduced, while no significant change in that of ATG9.

Fig. 1. Representative images of western blot of ATG7, ATG9, CD9 and GM130.

Reviewer #3 (Remarks to the Author):

Comment 1: I appreciate the efforts made by the authors to follow up all reviewer's suggestions. I only have a minor comment: I would suggest to include exosome treatment concentration in the legends of Fig. 4 and Supp. Fig. 4i), in the same way they did for Fig. 5c.

Answer: We thank the reviewer for his/her positive comments, addressing them has substantially improved our study. According to your suggestions, we have added the exosome treatment concentration in the legends of Fig. 4, Supp. Fig. 4i and Fig. 5c.

References

1. H. Guo, M. Chitiprolu, L. Roncevic, C. Javalet, F. J. Hemming, M. T. Trung, L. Meng, E. Latreille, C. Tanese de Souza, D. McCulloch et al., Atg5 Disassociates the V(1)V(0)-ATPase to Promote Exosome Production and Tumor Metastasis Independent of Canonical Macroautophagy. *Dev. Cell* 43, 716-730 e717 (2017).
2. M. Burke, W. Choksawangkar, N. Edwards, S. O. Rosenberg, C. Fenselau, Exosomes from myeloid-derived suppressor cells carry biologically active proteins. *J Proteome Res* 13, 836-843 (2014).
3. L. Gangoda, M. Liem, C. S. Ang, S. Keerthikumar, C. G. Adda, B. S. Parker, S. Mathivanan, Proteomic Profiling of Exosomes Secreted by Breast Cancer Cells with Varying Metastatic Potential. *Proteomics* 17, 23-24 (2017).
4. S. Anand, N. Foot, C. S. Ang, K. M. Gembus, S. Keerthikumar, C. G. Adda, S. Mathivanan, S. Kumar, Arrestin-Domain Containing Protein 1 (*Arrdc1*) Regulates the Protein Cargo and Release of Extracellular Vesicles. *Proteomics* 18, e1800266 (2018).
5. C. A. Bader, T. Shandala, Y. S. Ng, I. R. Johnson, D. A. Brooks, Atg9 is required for intraluminal vesicles in amphisomes and autolysosomes. *Biol Open* 4, 1345-1355 (2015).